

# Extracting statistically significant eddy signals from large Lagrangian datasets using wavelet ridge analysis, with application to the Gulf of Mexico

Jonathan M. Lilly[1] and Paula Pérez-Brunius[2]

[1]Theiss Research, La Jolla, California, USA
[2]Departamento de Oceanografía, Centro de Investigación Científica y de Educación Superior de Ensenada (CICESE), Ensenada, Mexico

**Correspondence:** Jonathan Lilly (j.m.lilly@theissresearch.org)

**Abstract.** A method for objectively extracting the displacement signals associated with coherent eddies from Lagrangian trajectories is presented, refined, and applied to a large dataset of 3761 surface drifters from the Gulf of Mexico. The method, wavelet ridge analysis, is modified to exclude the possibility of features changing from rotating in the cyclonic sense to rotating in the anticyclonic sense or vice-versa, transitions that would be physically unrealistic for a coherent eddy. A means for formally assessing statistical significance is introduced, addressing the issue of 'false positives' arising by chance from an unstructured turbulent background, and opening the door to confident application of the method to very large datasets. Significance is measured in a two-dimensional parameter space by comparison with a stochastic dataset having statistical and spectral properties that match the original, but lacking organized oscillations due to eddies or waves. The application to the Gulf of Mexico reveals massive asymmetries between cyclones and anticyclones, with anticyclones dominating at radii larger than about 50 km, but an unexpectedly rich population of highly nonlinear cyclones dominating at smaller radii. Both the method and the Gulf of Mexico eddy dataset are made freely available to the community for use in future research.

## 1 Introduction

Trajectories from freely-drifting, or Lagrangian, instruments are one of the major windows into observing the ocean circulation. A perennial theme in oceanography is the study of long-lived vortex structures, also known as coherent eddies, and their role in influencing the large-scale circulation. On account of these two factors, an important data analysis problem is to be able to accurately, objectively[1] and automatically detect signals in Lagrangian data arising from instruments trapped in coherent eddies, and to use these extracted signals to estimate the physical properties of the eddies themselves.

Obtaining a satisfactory solution to this problem that could scale to very large datasets would enable a rigorous eddy census of the entire surface drifter dataset from NOAA's Global Drifter Program (Lumpkin and Pazos, 2007), currently consisting of about 25000 trajectories. It would also, significantly, permit one to compare an eddy census from real-world surface drifters

---

[1]The term 'objective' is used here in its conventional sense of not being shaped by personal opinion, that is, non-subjective. This is not to be confused with its more technical definition, used by e.g. Haller (2005), meaning invariant to variations of an observer's frame of reference.



with those from synthetic trajectories in high-resolution numerical models. This type of Lagrangian model/data intercomparison offers considerable potential for refining models, and also for querying the models to better understand what is being observed in data, but is not yet possible due to limitations of available analysis methods.

Solutions to this problem has been proposed by many researchers over the years, based on a variety of methods (e.g. Kirwan
et al., 1984, 1988; Armi et al., 1989; Flament et al., 2001; Testor and Gascard, 2003; Shoosmith et al., 2005; Lankhorst, 2006); see Dong et al. (2011) for a useful review. Proposed methods span a range from visual inspection or simple filtering, to groundedness in dynamical or statistical modeling. While these methods have been successful for studying small numbers of trajectories, their application to large datasets would be problematic. A particular hurdle is that there are often free parameters that must be tuned by the analyst to an individual timeseries. What is desired is an inverse method that begins with specifying
the type of signal object that one is looking for, that is deeply rooted in dynamical theory, that has as few free parameters as possible, that proceeds without operator intervention, and that has accompanying measures of significance and estimates of errors.

The methods described above are what one may refer to as 'eddy-extraction methods', with the goal being to isolate the eddy signal from the background. The spin parameter method (Veneziani et al., 2004, 2005a, b) represents a different idea.
This method postulates a stochastic model for Lagrangian trajectories, in which a tendency to systematically rotate is introduced to capture the effect of eddies. It has the advantage of being relatively straightforward to apply to large datasets, and has been successfully employed to examine aspects of eddy statistics in global (Griffa et al., 2008; Lumpkin, 2016) as well as regional (e.g. Cetina-Heredia et al., 2019) studies. The goal of the spin parameter method is to quantify the bulk effects of eddies on trajectories. It does not attempt a precise separation between the eddy displacements and everything else, and would be expected
to perform poorly at this task except under idealized conditions. This method is therefore different from, and complementary to, the eddy extraction methodology we are discussing here.

Another body of work that should be mentioned involves a recent breakthrough in the ability to objectively identify coherent eddies through Lagrangian analysis of the full two-dimensional flow field (Haller, 2005; Haller and Beron-Vera, 2012). This innovation has spawned a host of applications to numerical models (Karrasch et al., 2015) as well as to real-world data
(Beron-Vera et al., 2008, 2013; Abernathey and Haller, 2018), including several in the Gulf of Mexico (Olascoaga et al., 2013; Beron-Vera et al., 2018). It has inspired variations (e.g Conti et al., 2016) as well as a theoretical extension to three-dimensional flows (Blazevski and Haller, 2014). This method, while also based on a Lagrangian framework and also aimed at objectively identifying vortices, relates to a quite different problem from that considered here, namely identifying eddies from two-dimensional flow fields (e.g. McWilliams, 1990). In the present paper we are examining Lagrangian trajectories from in-
dividual instruments, without access to the entire two-dimensional flow fields that are required in the method of Haller (2005) and its relatives.

A major step in the eddy extraction problem was taken by Lilly and Gascard (2006). In that paper, an innovative and powerful method from signal analysis termed *wavelet ridge analysis* (Delprat et al., 1992; Mallat, 1999) was modified for application to Lagrangian trajectories. That method is designed to detect and analyze modulated oscillations, that is, oscillations whose
amplitude and frequency vary as a function of time. This type of signal accords well with our physical expectations for the





trajectory of a particle trapped in a vortex. In particular, experience shows that eddy-trapped trajectories exhibit substantial frequency modulation, implying that narrowband methods such as bandpassing or complex demodulation will perform poorly. The wavelet ridge method is able to automatically extract frequency-modulated signals from a timeseries with no analyst intervention, subject to only a handful of free parameters. A compelling aspect of this method is that it begins with the specifi-
cation of the type of object we are looking for, namely, a modulated oscillation, a type of signal for which there exists a solid theoretical foundation (Gabor, 1946; Picinbono, 1997).

An application of the wavelet ridge analysis method to trajectories from a numerical model of an unstable jet on a beta plane (Lilly et al., 2011) showed that the method could accurately and automatically extract eddy signals from a set of 100 trajectories, and that these could be unambiguously discriminated from oscillatory signals due to Rossby waves or jet meander.
The wavelet ridge analysis method has been employed by a number of other authors (e.g. Garreau et al., 2011; Alpers et al., 2013; Bower et al., 2013; Le Hénaff et al., 2014; Inoue et al., 2016; Kourafalou et al., 2017; Furey et al., 2018; de Jong et al., 2018; Le Hénaff et al., 2020) using the software implementation created by Lilly and Gascard (2006), and appears to now be the standard method for extracting eddy signals from Lagrangian data.

A series of advances followed the prototype method presented in Lilly and Gascard (2006). The mathematical model of a
univariate modulated oscillation was extended to two dimensions by Lilly and Olhede (2010b), who showed that such a signal is rightly thought of as the trajectory traced out by a particle orbiting an ellipse with time-varying properties, in other words, a modulated elliptical signal. The same interpretation holds in three dimensions (Lilly, 2011), where the plane containing the ellipse can also vary its orientation in time. Wavelet ridge analysis was generalized to multidimensional signals by Lilly and Olhede (2009a) and Lilly and Olhede (2012a). Finally, estimates for bias errors arising from modulation strength were
rigorously found through a perturbation analysis for both the univariate (Lilly and Olhede, 2010a) and multivariate cases (Lilly and Olhede, 2012a).

The wavelet ridge analysis method allows one to readily analyze datasets with dozens or perhaps hundreds of trajectories. However there is a major challenge which prevents it—or any other eddy-extraction method—from being immediately applicable to very large datasets consisting of thousands or tens of thousands of trajectories. This is the problem of 'false positive'
features arising from the interaction of the detection method with the turbulent background flow. For small to medium-size datasets such events may be readily identified with a visual scan, but that subjective operation becomes unwieldy for larger datasets.

The problem of false positives can be understood as follows. Trajectories in the ocean's background of geostrophic turbulence can be idealized as resulting from a type of damped random walk, see Lilly et al. (2017). For the one-dimensional case, a
discrete random walk is intuitively described as a drunk staggering between lampposts. For the two-dimensional case, the drunk has a grid of lampposts available for their staggering. From time to time the drunk will, by chance, happen to turn in a circle, or oscillate back and forth between two lampposts. This illustrates why, in applying the wavelet ridge analysis to timeseries of stochastic processes analogous to the random walk, oscillatory events are occasionally detected. One would not wish to confuse random features arising from the turbulent background with the organized oscillations due to coherent eddies.


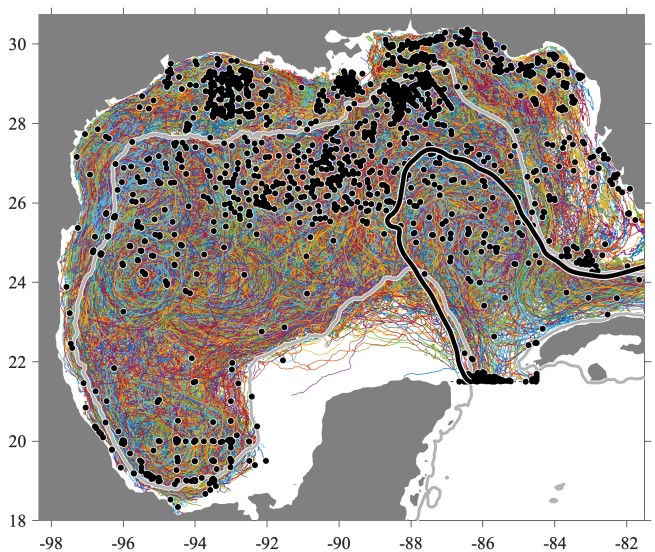

**Figure 1.** Surface drifter trajectories in the Gulf of Mexico from the consolidated dataset of Lilly and Pérez-Brunius (2020a). Colors correspond to different trajectories, the beginning of each of which is marked by a black dot. In this and the following plot, the heavy gray contour is the 500 m isobath, while the heavy black line outlines the estimated location of the time-mean Loop Current. Specifically, the black contour marks the location at which the drifter-estimated time-mean speed from Fig. 3a of Lilly and Pérez-Brunius (2020a) is equal to 70 cm s$^{-1}$. The southern edge of the study region in this figure and in Fig. 2 is delineated by the dotted line extending east from the Yucatán Peninsula along 21.5° N, then turning north to Cuba at 84.5° W.

This paper has three objectives. Firstly, the ridge analysis method of Lilly and Gascard (2006) is streamlined and updated with several important refinements, a deeper discussion of the connection to signal processing theory, and with sufficient background on the wavelet transform itself so as to be self-contained. Secondly, the method is applied to a set of several thousand drifter trajectories from the Gulf of Mexico, see Fig. 1, leading to the identification of a set of statistically significant eddy signals, shown in Fig. 2. This Gulf of Mexico eddy dataset is freely distributed to the community, as described at the end

of the paper, and is being used in a sequel to investigate the eddy dynamics in this environmentally and societally important region.

     Finally, the false-positive problem is solved through (i) the creation of a 'noise' dataset, matching the basic statistical properties of the observations but lacking the organized oscillatory structures associated with eddies, together with (ii) the introduction of an innovative two-dimensional significance criterion. The noise dataset encodes the null hypothesis that the observations

result from an isotropic, eddyless random flow. Detected signals are judged to be statistically significant if their duration—measured by the number of successive oscillations—and/or degree of circular polarization are sufficiently distinguished from that expected due to the noise. Importantly, this criterion makes no explicit assumptions regarding the size, velocity, or degree of nonlinearity of the features we seek to detect. In other words, the criterion is based on the essential properties of a coher-

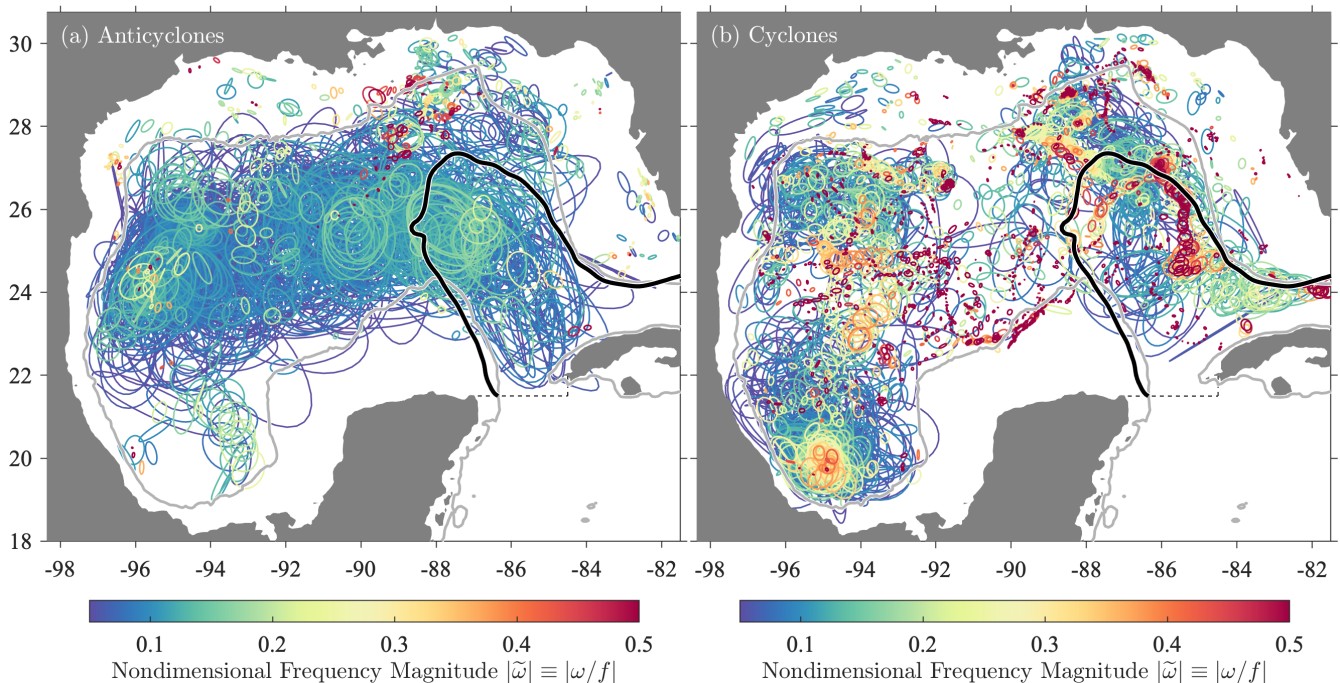

**Figure 2.** Ellipses from all statistically significant eddy signals in the dataset shown in Fig. 1 detected using the method created here, with anticyclonic events in (a) and cyclonic events in (b). The time interval between successive ellipses is equal to the estimated period of the oscillation, $2\pi/|\omega(t)|$, with the instantaneous frequency $\omega(t)$ defined later in the text. The colored shading is nodimensional instantaneous frequency magnitude $|\widetilde{\omega}(t)| \equiv |\omega(t)/f_\diamond(t)|$, in which $\omega(t)$ has been normalized by the Coriolis frequency $f_\diamond(t)$ at the latitude of the ellipse's center. It will be shown that $2|\widetilde{\omega}(t)|$ is an estimate of the local Rossby number of the oscillatory flow. Ellipses are plotted in order of their $|\widetilde{\omega}(t)|$ value, thus placing higher-frequency, more nonlinear features on top. The asymmetry between cyclonic and anticyclonic features is strikingly apparent. Other lines are as in Fig. 1.

ent eddy, namely being long-lived and nearly circular, but is agnostic to specific details such as size, strength, or circulation

timescale.

As motivation for the investment required in learning this method, the results of the application to the Gulf of Mexico will be briefly described at the outset. The dataset to be analyzed in this work is a set of 3761 surface drifter trajectories, compiled from a variety of experiments, processed, and quality controlled as described in Lilly and Pérez-Brunius (2020a), shown in Fig. 1. Applying the method as described here, one is able to extract from the drifter trajectories a set of contiguous, possibly

overlapping displacement signals consisting of modulated oscillations in two dimensions. The portion of such signals judged to be statistically significant, in the sense of occurring at a rate at least $10\times$ greater than that expected under the noise hypothesis, is identified. These are visualized in Fig. 2 as a series of ellipses, colored here by a nondimensional frequency magnitude that when multiplied by two becomes a measure of Rossby number.





The distributions of cyclonic and anticyclonic events are completely different. The dense populations of large anticyclonic

eddies filling the central Gulf are, naturally, the well-known Loop Current eddies (e.g Elliott, 1982; Lipphardt Jr. et al., 2008; Hall and Leben, 2016). These shed periodically from the energetic Loop Current on the Gulf's eastern side, then propagate westward. Only a handful of small, intense anticyclones are observed, mostly located between the Loop Current and the Mississippi outflow region. On the cyclonic side, a hotspot of activity is seen in the southern Gulf, corresponding to a persistent cyclonic eddy known as the Campeche Gyre (Padilla-Pilotze, 1990; Vázquez De La Cerda et al., 2005; Pérez-Brunius et al.,

2013). On the periphery of the Loop Current, a population of intense mesoscale eddies are seen, the Loop Current Frontal Eddies (LCFEs), see Le Hénaff et al. (2014) and references therein. Similar features occur also in the western Gulf but are of an unknown origin. In what appears to be a new result, sub-10 km, submesoscale eddies with Rossby numbers approaching or exceeding unity are found throughout the Gulf, visible in this plot as small red circles. This figure is an unprecedented view of the eddy activity in the Gulf, and illustrates the potential of the method to illuminate aspects of the circulation that are

otherwise buried in trajectories.

The structure of the paper is as follows. A mathematical model for the motion of a particle trapped in an eddy is presented in Sect. 2. The means of extracting such signals from displacement timeseries that are also influenced by a turbulent background flow is developed in Sect. 3, building on Lilly and Gascard (2006) and subsequent work. The application to the Gulf of Mexico and the creation of a statistical significance test are accomplished in Sect. 4. A brief discussion of the statistically

significant eddy events is given, followed by the Conclusions. Further examination of the Gulf of Mexico eddy signals and their implications for eddy dynamics and life cycles in this region is left to a sequel.All code developed for this paper is distributed as a part of a freely available Matlab toolbox, including a convenient and self-contained eddy extraction function, as described following the Conclusions. Finally the Gulf of Mexico Eddy Dataset (GOMED), including the signals shown in Fig. 2 and their associated ellipse parameters, is described. Because a large number of different mathematical symbols are used

in this paper, Table A1, located after the conclusions, presents an overview of some of the most important ones.

## 2 A model for particle motion in an eddy

This section describes a mathematical method for the displacement signal of a particle trapped in coherent eddy, building on that formulated by Lilly and Gascard (2006), Lilly et al. (2011), and Lilly and Olhede (2012a).

### 2.1 The modulated elliptical signal

The displacement signal of an instrument or particle advected by an eddy, within a moving frame of reference centered on the eddy's center, will be modeled as the trajectory traced out by a particle orbiting a time-varying ellipse. We will use the complex-valued notation $z(t) = x(t) + \mathrm{i}y(t)$, with $x(t)$ and $y(t)$ being the east-west and north-south displacements from the eddy center, respectively. Such a signal can be written as

$$z(t) \equiv \mathrm{e}^{\mathrm{i}\theta(t)}\left[a(t)\cos\phi(t) + \mathrm{i}b(t)\sin\phi(t)\right], \tag{1}$$





where $a(t)$ and $b(t)$ are the semi-major and semi-minor axes, $\theta(t)$ is the orientation angle of the major axis with respect to the $x$-axis, and $\phi(t)$ is a phase angle setting the position of the particle along the ellipse periphery. While $a(t) \geq |b(t)|$ is always positive, $b(t)$ can be of either sign and encodes the direction in which the ellipse is being orbited, with positive $b(t)$ corresponding to counterclockwise motion. Eq. (1) will be called the *ellipse generation equation.*

A schematic of an ellipse is presented in Fig. 3. The geometric angle between the major axis and the particle position, denoted $\varphi(t)$ in this plot, is related to the phase $\phi(t)$ that appears in Eq. (1) by[2]

$$\tan\varphi(t) = \frac{b(t)}{a(t)}\tan\phi(t) \tag{2}$$

such that $\varphi(t)$ can be recovered from the phase $\phi(t)$ via

$$\varphi(t) = \Im\left\{\log\left[a(t)\cos\phi(t) + ib(t)\sin\phi(t)\right]\right\} \tag{3}$$

where $\Im\{\cdot\}$ denotes the imaginary part. The combination $\Im\{\log(\cdot)\}$ is used here to implement the four-quadrant inverse tangent function, since $\Im\left\{\log e^{i\vartheta}\right\} = \vartheta$.

The type of signal described by Eq. (1) is referred to as a *modulated elliptical signal* (Lilly and Gascard, 2006; Lilly and Olhede, 2010b). A special case of the modulated elliptical signal is one in which the modulation vanishes, that is, a purely sinusoidal oscillation in two dimensions. With the phase linearly increasing at some fixed rate $\omega_o > 0$, $\phi(t) = \omega_o t + \phi_o$, and with the other three ellipse parameters being the constants $a_o$, $b_o$, and $\theta_o$, Eq. (1) becomes

$$z_o(t) \equiv e^{i\theta_o}\left[a_o\cos(\omega_o t + \phi_o) + ib_o\sin(\omega_o t + \phi_o)\right] \tag{4}$$

which describes the sinusoidal orbiting of a fixed ellipse. The motion is purely circular for $a_o = |b_o|$ and rectilinear for $b_o = 0$. (Note that $\omega_o$ is chosen to be positive because the convention is adopted that the sign of $b_o$ sets the direction of rotation.) In general the modulated oscillatory signal $z(t)$ differs from a pure sinusoid in that the properties of the ellipse are allowed to change in time. Thus, the modulated elliptical signal is a generalization of an AM/FM or amplitude modulated / frequency modulated signal to two dimensions.

The ellipse generation equation of Eq. (1) is purely kinematic, yet it accords well with our physical understanding of potential paths of particles trapped in coherent vortices. To begin with, even if a vortex is circular, its properties may change with time through, for example, geostrophic adjustment. Moreover, the *observed* vortex properties may change as the measuring instrument shifts position within the vortex to a new radius. This may be due to slight non-Lagrangian behavior of the instrument, or to wind-driven motion of the surface layer that is independent from the vortex motion. Thus $z(t)$ includes modulation both due to time variation of the vortex itself, as well due to a possible profiling effect from an instrument drifting within a fixed vortex.

While oceanic vortices are often nearly circular, there are several reasons to permit elliptical motion in our conceptual model. Material ellipses arise naturally in considering a second-order Taylor expansion of a two-dimensional flow (e.g. Kirwan et al.,

---

[2]The lead author thanks S. Elipot for pointing out an error in earlier published versions of Fig. 3, in which the angle labeled $\varphi$ here was incorrectly identified as $\phi$.


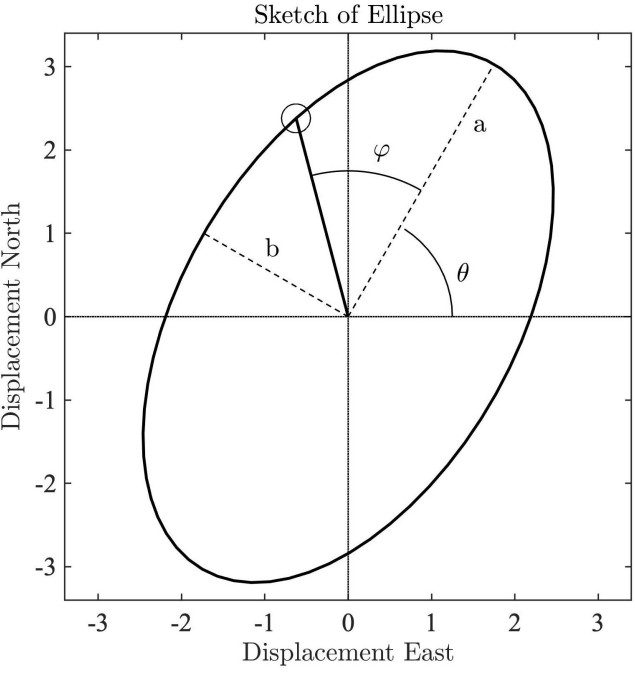

**Figure 3.** A schematic of a particle orbiting an ellipse, as described by Eq. (1). The semi-major axis $a(t)$, semi-minor axis $b(t)$, orientation angle $\theta(t)$, and particle position angle $\varphi(t)$ are all labeled. The circle denotes the instantaneous particle position. Here $a(t) = 3.5$, $b(t) = 2$, and $\theta(t) = \phi(t) = \pi/3$. These values imply $\varphi(t) = \pi/4.03$ together with $\kappa(t) \equiv \sqrt{[a^2(t) + b^2(t)]/2} = 2.85$ and $\varsigma(t) \equiv 2a(t)b(t)/[a^2(t) + b^2(t)] = 0.93$, with the latter two quantities defined subsequently in Sect. 2.2.

1984; Lilly, 2018). Moreover, in some cases the vortex itself may be elliptical. In quasigeostrophic dynamics, ellipticization is seen as a natural response of a circular vortex to ambient shear or strain (e.g. Ruddick, 1987; Meacham and Flierl, 1990; McKiver and Dritschel, 2006), while a freely evolving, unforced elliptical vortex patch under shallow water dynamics is known to be a good model for Gulf Stream Rings and other similar large eddies (e.g. Cushman-Roisin et al., 1985; Young, 1986; Ripa, 1987). A more detailed review of elliptical vortex solutions and their geophysical applications may be found in Lilly (2018).

Similarly, the trajectory of a particle entraining into or detraining from a vortex in a realistic flow may appear as an elliptically polarized oscillation, even if the vortex itself is circular. For all of these reasons the modulated elliptical signal is preferable to a modulated oscillation having a purely circular polarization, i.e. with $|b(t)| = a(t)$.

The advection of a particle by a coherent eddy is not, however, the only type of physical phenomenon giving rise to a modulated elliptical signal. This model also matches the displacement signal expected for waves, most notably inertial oscillations.

These are expected to have an anticyclonic circular polarization, i.e. $b(t) = -a(t)$ in the Northern Hemisphere and $b(t) = a(t)$ in the Southern Hemisphere. Therefore, a method designed to detect eddies will also detect inertial oscillations. As discussed earlier, a background stochastic process—such as is expected due to position fluctuations arising from geostrophic turbulence, for example—may, by chance, also lead to some signals that are well described by Eq. (1). In identifying modulated elliptical





signals with the goal of studying eddies, one must take care to account for the possibility of such false positives; this is done in
Sect. 4.

## 2.2  Alternate ellipse parameters

It proves convenient to replace the semi-axes lengths $a(t)$ and $b(t)$ with measures of the ellipse size and shape,

$$\kappa^2(t) \equiv \frac{a^2(t) + b^2(t)}{2}, \qquad \varsigma(t) \equiv \frac{2a(t)b(t)}{a^2(t) + b^2(t)} \tag{5}$$

where $\kappa(t)$ is the root-mean-square axis length, and $\varsigma(t)$ is a signed quantity termed the *circularity*. Note that $|\varsigma(t)| \leq 1$, with
$\varsigma(t) = 1$ for positive circular oscillations, $\varsigma(t) = -1$ for negative circular oscillations, and $\varsigma(t) = 0$ for linear oscillations. Thus,
highly eccentric or elongated ellipses have small degrees of circularity.[3]

The ellipse generation equation, Eq. (1), can be rewritten in terms of two counter-rotating circular signals as

$$z(t) = a^+(t)e^{i\phi^+(t)} + a^-(t)e^{-i\phi^-(t)} \tag{7}$$

where these "rotary" amplitudes and phases are given in terms of the ellipse parameters as (Lilly and Gascard, 2006)

$$a^+(t) \equiv \frac{1}{2}\left[a(t) + b(t)\right], \qquad \phi^+(t) \equiv \phi(t) + \theta(t) \tag{8}$$

$$a^-(t) \equiv \frac{1}{2}\left[a(t) - b(t)\right], \qquad \phi^-(t) \equiv \phi(t) - \theta(t). \tag{9}$$

Substituting $a^+(t)$ and $a^-(t)$ for $a(t)$ and $b(t)$, one finds the circularity can be alternately expressed as

$$\varsigma(t) = \frac{2a(t)b(t)}{a^2(t) + b^2(t)} = \frac{\left[a^+(t)\right]^2 - \left[a^-(t)\right]^2}{\left[a^+(t)\right]^2 + \left[a^-(t)\right]^2} \tag{10}$$

which reveals it to be an instantaneous version of the *rotary coefficient* introduced by Gonella (1971), a fundamental measure
of the normalized difference between positively-rotating and negatively-rotating signal energy.

## 2.3  The analytic signal

The ellipse generation equation, Eq. (1), is understood to mean that if the time-varying ellipse properties $a(t)$, $b(t)$, $\theta(t)$, and
$\phi(t)$ on the right-hand side are prescribed, the oscillatory displacement signal $z(t)$ emerges as a result. However, the observa-
tional problem, in the simplest case of a single modulated oscillation and vanishing background flow, is to work backwards
from an observed modulated oscillation $z(t)$ to determine the unobserved ellipse parameters that created it. At first glance this
seems impossible, because $z(t)$ consists of two real-valued numbers at each time and we are interested in recovering four such

---

[3]The circularity is related to another measure of ellipse shape, the *linearity* used by Lilly and Gascard (2006) and Lilly and Olhede (2010b), defined as

$$\lambda(t) = \text{sgn}\left(b(t)\right)\frac{a^2(t) - b^2(t)}{a^2(t) + b^2(t)} \tag{6}$$

where $\text{sgn}(\cdot)$ is the signum function defined in Eq. (18). The linearity and circularity are related through $\lambda^2(t) + \varsigma^2(t) = 1$.



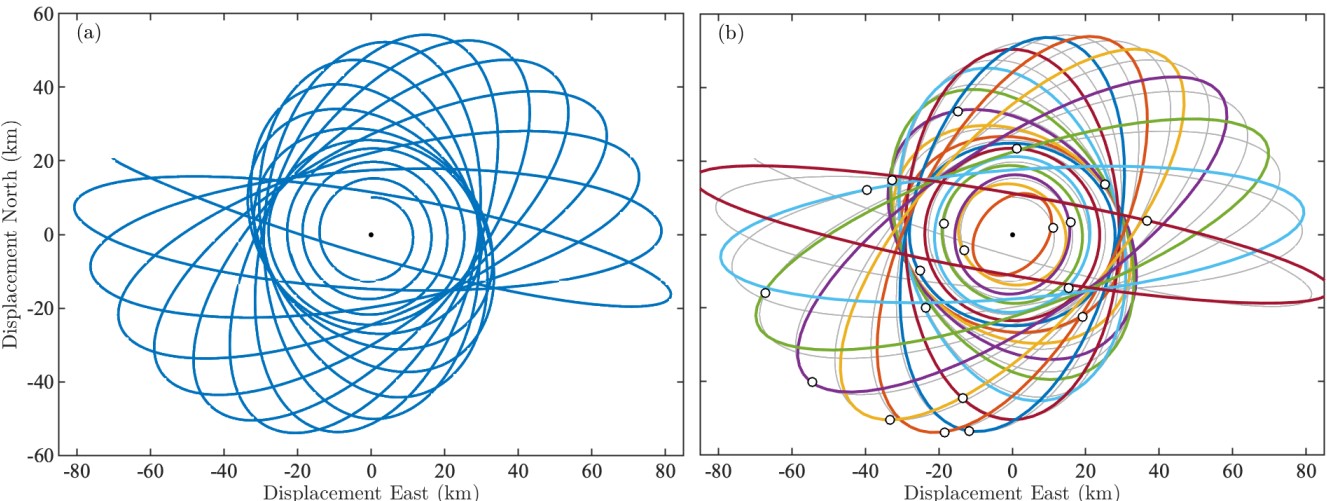

**Figure 4.** A synthetically constructed modulated elliptical signal (a) together with (b) ellipses inferred from the analytic signal method. See text for details. The thin gray line in panel (b) is the same as the blue line in panel (a). Dots in panel (b) gives the state of the signal at the moments at which the ellipses are shown; these are intersection points of the signal curve and the ellipses.

numbers. However, this inverse problem does indeed have a meaningful unique solution using the so-called *analytic signal method*, provided one adds some additional constraints.

It is useful at this point to show an example. The signal in Fig. 4a is synthetically constructed from the ellipse generation equation, Eq. (1), with parameter choices given in Appendix A. This anticyclonically-orbiting signal starts out circular for the first quarter of its duration, then transitions to a uniformly precessing ellipse ($\theta'(t) = \text{constant}$), with an increasing eccentricity or decreasing circularity $|\varsigma(t)|$, all the while linearly growing in amplitude $\kappa(t)$ and also, though not apparent in this plot, in frequency. Using the analytic signal method discussed subsequently, one may assign a time-varying ellipse to this signal, snapshots of which are shown in Fig. 4b every 50 days beginning on the 25th day of this 1000 day timeseries. The ellipse properties so inferred, when themselves inserted into the generation equation, exactly reproduce the original signal.

To develop this method, we first turn to the case of a univariate signal. The modulated elliptical signal is a generalization to two dimensions of an amplitude-modulated and frequency-modulated univariate oscillation

$$x(t) \equiv a_x(t) \cos \phi_x(t) \tag{11}$$

where $a_x(t) > 0$ is a time-varying, or *instantaneous*, amplitude and $\phi_x(t)$ is an instantaneous phase. Signals of this type include pure sinusoids as a special case, with $a_x(t) = a_o$ and $\phi_x(t) = \omega_o t + \phi_o$, but are far more general. A simple example of a modulated oscillation is a wave packet. A wave packet exhibits amplitude modulation in the form of its envelope, and possibly frequency modulation if the frequency content is not uniform over time. Nevertheless, it is also distinctly oscillatory. Eq. (11) captures the essence of such oscillatory features that depart from being strictly sinusoidal.





Given $x(t)$, one wishes to recover the $a_x(t)$ and $\phi_x(t)$ that could generate it via Eq. (11). That is, we wish to associate with $x(t)$ an amplitude $a_x(t)$ and a phase $\phi_x(t)$, in order to conceptualize and describe $x(t)$ as a modulated oscillation. The association of a physically meaningful amplitude and phase with a given signal is an underdetermined problem, as an infinite family of amplitude/phase pairs on the right-hand side can generate the observed signal on the left-hand side. However, a meaningful solution to this problem was found by Gabor (1946) in a landmark paper, by introducing what is now known as the *analytic signal* method, see e.g. Picinbono (1997) and references therein for a more modern treatment.

In the analytic signal method, a particular amplitude/phase pair, known as the *canonical* amplitude and phase, are determined by pairing the real-valued signal $x(t)$ with an imaginary part consisting of its own Hilbert transform,

$$a_x(t)e^{i\phi_x(t)} \equiv x^+(t) \equiv [1 + i\mathcal{H}]\,x(t) \tag{12}$$

where $\mathcal{H}$ is the Hilbert transform operator defined as

$$\mathcal{H}\{x(t)\} \equiv \frac{1}{\pi}\fint_{-\infty}^{\infty} \frac{x(u)}{t-u}\,\mathrm{d}u \tag{13}$$

with "$\fint$" being the Cauchy principal value integral. The resulting complex-valued signal $x^+(t)$, the real part of which is the same as the original signal, $x(t) = \Re\{x^+(t)\}$, is called the *analytic signal* associated with $x(t)$, or its *analytic part*.

The canonical amplitude is defined uniquely in terms of the analytic signal by $a_x(t) \equiv |x^+(t)|$, and the canonical phase function by $\phi_x(t) \equiv \Im\{\log(x^+(t)/|x^+(t)|)\}$. For the special case of a sinusoidal signal $x(t) = a_o\cos(\omega_o t + \phi_o)$, the analytic signal is given by, as will be shown shortly,

$$x^+(t) \equiv [1 + i\mathcal{H}]\,a_o\cos(\omega_o t + \phi_o) = a_o e^{i\omega_o t + i\phi_o}. \tag{14}$$

Thus for a sinusoid, the amplitude and phase assigned using the analytic signal method agree with the obvious choices, i.e. $a_x(t) = a_o$ and $\phi_x(t) = \omega_o t + \phi_o$. This crucial property is known as *harmonic correspondence* (Vakman, 1996). The canonical phase can be differentiated to give the *instantaneous frequency* (Gabor, 1946; Picinbono, 1997),

$$\omega_x(t) \equiv \phi_x'(t), \tag{15}$$

where the prime denotes a time derivative. The instantaneous frequency allows for the objective quantification of the time-varying frequency content of a modulated oscillation. See Lilly and Olhede (2010b) and references therein for a discussion of the close connection between the instantaneous frequency and the conventional definition of a mean frequency from first moment of $|X(\omega)|^2$ in the Fourier domain.

## 2.4 Analytic signal details

To better understand the analytic signal, we first note that the Hilbert transform has a simple action in the frequency domain. Let $X(\omega)$ be the Fourier transform of $x(t)$, from which $x(t)$ is recovered using the inverse Fourier transform

$$x(t) = \frac{1}{2\pi}\int_{-\infty}^{\infty} X(\omega)e^{i\omega t}\mathrm{d}\omega. \tag{16}$$





One may show that the Hilbert transform of $x(t)$ is given by (e.g. Papoulis, 1962, pp 37–38)

$$\mathcal{H}\{x(t)\} = \frac{-\mathrm{i}}{2\pi} \int\limits_{-\infty}^{\infty} \mathrm{sgn}(\omega) X(\omega) \mathrm{e}^{\mathrm{i}\omega t} \mathrm{d}\omega \tag{17}$$

where the signum function

$$\mathrm{sgn}\,(x) \equiv \begin{cases} 1 & x > 0 \\ 0 & x = 0 \\ -1 & x < 0 \end{cases} \tag{18}$$

returns the sign of its argument. The Hilbert transform thus simply changes the sign of Fourier coefficients for negative frequencies, while lagging all Fourier phases by ninety degrees. It follows that the analytic signal can be written as

$$[1 + \mathrm{i}\mathcal{H}]\,x(t) = \frac{1}{2\pi} \int\limits_{-\infty}^{\infty} 2U(\omega)X(\omega)\mathrm{e}^{\mathrm{i}\omega t}\mathrm{d}\omega \tag{19}$$

where $U(x) \equiv \frac{1}{2}\left[1 + \mathrm{sgn}(x)\right]$ is the unit step function. Thus the analytic operator $[1 + \mathrm{i}\mathcal{H}]$ has the action of doubling the Fourier coefficients of all positive frequencies, while setting those of negative frequencies to zero. A signal whose Fourier transform vanishes at negative frequencies is said to be *analytic*.

Recall that a cosine and sine have the Fourier representations of

$$\cos(\omega_o t) = \frac{1}{2\pi} \int\limits_{-\infty}^{\infty} \pi\left[\delta(\omega - \omega_o) + \delta(\omega + \omega_o)\right]\mathrm{e}^{\mathrm{i}\omega t}\mathrm{d}\omega \tag{20}$$

$$\sin(\omega_o t) = \frac{-\mathrm{i}}{2\pi} \int\limits_{-\infty}^{\infty} \pi\left[\delta(\omega - \omega_o) - \delta(\omega + \omega_o)\right]\mathrm{e}^{\mathrm{i}\omega t}\mathrm{d}\omega \tag{21}$$

where $\delta(\omega)$ is the Dirac delta function. From these, Eq. (17) shows that the action of the Hilbert transform is to decrement the phases of all sinusoidal components by ninety degrees, i.e. $\mathcal{H}\{\cos(\omega_o t)\} = \sin(\omega_o t)$ and $\mathcal{H}\{\sin(\omega_o t)\} = -\cos(\omega_o t)$. Harmonic correspondence, Eq. (14), follows.

A compelling argument in favor of the analytic signal method is due to Vakman (1996); see also Huang and Yang (2011).
Consider assigning a time-varying amplitude and phase by pairing the real-valued signal $x(t)$ with some other imaginary part than its Hilbert transform. That is, in Eq. (12) we replace $[1 + \mathrm{i}\mathcal{H}]$ with $[1 + \mathrm{i}\mathcal{L}]$ where $\mathcal{L}$ is some yet-to-be-determined linear operator. Vakman proves that the Hilbert transform $\mathcal{H}$ is the only linear operator that leads to an amplitude and phase pair with the following three properties: (i) harmonic correspondence; (ii) *amplitude continuity*, meaning that small variations $\delta x(t)$ of the signal $x(t)$ should correspond to small variations $\delta a_x(t)$ of the amplitude; and (iii) *phase invariance to scaling*, such that
the instantaneous phase of $x(t)$ and that of a rescaled signal $cx(t)$, where $c > 0$, should be identical. Vakman's conditions leave little choice but to accept that the analytic signal method is the natural way to solve this inverse problem.

Note that there are actually two different sets of amplitude/phase parameters: those used to *generate* a signal via the right-hand side of Eq. (11), which are unobservable, and the canonical amplitude and phase that are *assigned* to an observed signal





using Eq. (12). Here we have used the same symbols to denote both sets in order to avoid a cumbersome notation. While both
sets of parameters generate the same signal, and while they are guaranteed to be identical for the case of a sinusoidal signal by
harmonic correspondence, in general, the two sets of parameters need not be the same.

A condition for when the generating amplitude and phase are identical to the canonical amplitude and phase was found in a
remarkable paper by Bedrosian (1963). The so-called *Bedrosian condition* for a real-valued univariate signal can be viewed as
a type of slow variation condition, in which the frequency-domain support for the amplitude function $a_x(t)$ exists on strictly
lower (that is, smaller magnitude) frequencies than the support of $\cos \phi_x(t)$. Broadly speaking, the difference between the
generation parameters and the canonical parameters is expected to become larger as the strength of modulation specified by
the generation parameters increases.

## 2.5 Inferring ellipse properties

The analytic signal method of assigning a time-varying amplitude and phase to a univariate signal $x(t)$ was generalized to the
modulated elliptical signal of Eq. (1) by Lilly and Olhede (2010b), which we will simplify and build on in this section. In
vector notation, the ellipse generation equation becomes

$$
\mathbf{x}(t) = \begin{bmatrix} x(t) \\ y(t) \end{bmatrix} = \mathbf{R}\left(\theta(t)\right) \begin{bmatrix} a(t)\cos\phi(t) \\ b(t)\sin\phi(t) \end{bmatrix}
\tag{22}
$$

where $\mathbf{R}(\vartheta)$ is the rotation matrix through some angle $\vartheta$,

$$
\mathbf{R}(\vartheta) \equiv \begin{bmatrix} \cos\vartheta & -\sin\vartheta \\ \sin\vartheta & \cos\vartheta \end{bmatrix}.
\tag{23}
$$

The analytic version of the bivariate signal $\mathbf{x}(t)$ is

$$
\mathbf{x}^+(t) \equiv [1 + \mathrm{i}\mathcal{H}]\mathbf{x}(t)
\tag{24}
$$

in terms of which the canonical ellipse parameters are defined via

$$
\mathrm{e}^{\mathrm{i}\phi(t)}\mathbf{R}\left(\theta(t)\right) \begin{bmatrix} a(t) \\ -\mathrm{i}b(t) \end{bmatrix} \equiv \mathbf{x}^+(t).
\tag{25}
$$

This assignment is readily verified to satisfy harmonic correspondence for the special case of a sinusoidal ellipse with fixed
geometry, that is, with $a(t) = a_o$, $b(t) = b_o$, $\theta(t) = \theta_o$, and $\phi(t) = \omega_o t + \phi_o$.

To infer the ellipse parameters from $\mathbf{x}^+(t)$, it is convenient to use a set of matrices introduced by Lilly (2018). Define

$$
\mathbf{J} \equiv \begin{bmatrix} 0 & -1 \\ 1 & 0 \end{bmatrix}, \qquad \mathbf{K} \equiv \begin{bmatrix} 1 & 0 \\ 0 & -1 \end{bmatrix}, \qquad \mathbf{L} \equiv \begin{bmatrix} 0 & 1 \\ 1 & 0 \end{bmatrix}
\tag{26}
$$





where $\mathbf{J} = \mathbf{R}\left(\pi/2\right)$ is recognized as the ninety-degree counterclockwise rotation matrix, and $\mathbf{K}$ and $\mathbf{L}$ are the reflection matrices about the lines $y = 0$ and $x = y$, respectively. $\mathbf{K}$ and $\mathbf{L}$ are found to transform under rotations as

$$\mathbf{R}^T(\theta)\mathbf{K}\mathbf{R}(\theta) = \cos 2\theta\,\mathbf{K} - \sin 2\theta\,\mathbf{L} \tag{27}$$

$$\mathbf{R}^T(\theta)\mathbf{L}\mathbf{R}(\theta) = \cos 2\theta\,\mathbf{L} + \sin 2\theta\,\mathbf{K} \tag{28}$$

as shown in Lilly (2018). Here the superscript "$T$" denotes the matrix transpose.

Using for convenience $\kappa(t)$ and $\varsigma(t)$ rather than $a(t)$ and $b(t)$, one finds that the definition in Eq. (25) implies

$$\kappa^2(t) = \frac{1}{2}\|\mathbf{x}^+(t)\|^2 \tag{29}$$

$$\varsigma(t) \equiv \frac{\Im\left\{\mathbf{x}^{+H}(t)\mathbf{J}\mathbf{x}^+(t)\right\}}{\|\mathbf{x}^+(t)\|^2} \tag{30}$$

$$\theta(t) \equiv \frac{1}{2}\Im\left\{\log\left[\mathbf{x}^{+H}(t)\mathbf{K}\mathbf{x}^+(t) + i\mathbf{x}^{+H}(t)\mathbf{L}\mathbf{x}^+(t)\right]\right\} \tag{31}$$

$$\phi(t) \equiv \frac{1}{2}\Im\left\{\log\left[\mathbf{x}^{+T}(t)\mathbf{x}^+(t)\right]\right\} \tag{32}$$

for the values of the canonical ellipse parameters expressed in terms of the analytic signal $\mathbf{x}^+(t)$. Here "$H$" is the Hermitian or conjugate transpose, $\|\mathbf{x}\| \equiv \sqrt{\mathbf{x}^H\mathbf{x}}$ is the norm of some vector $\mathbf{x}$, and we have made use of the rotation formulas, Eqs. (27) and (28), in the expression for $\theta(t)$. Finally

$$a(t) = \kappa(t)\sqrt{1 + \sqrt{1 - \varsigma^2(t)}} \tag{33}$$

$$b(t) = \kappa(t)\sqrt{1 - \sqrt{1 - \varsigma^2(t)}} \times \mathrm{sgn}\left(\varsigma(t)\right) \tag{34}$$

recovers the semi-axes lengths from $\kappa(t)$ and $\varsigma(t)$, as are obtained by inverting Eq. (5).

Equations (29)–(32) will be called the *ellipse inversion equations*, in the sense of going backwards from an observed signal to ellipse parameters. These expressions are more direct than those given in Lilly and Gascard (2006) and Lilly and Olhede (2010b), which use the rotary amplitudes and phases as an intermediate step in the inversion. Returning to the synthetic example, grouping the $x(t)$ and $y(t)$ signals shown in Fig. 4 into a vector $\mathbf{x}(t) = [x(t)\ y(t)]^T$, taking its analytic part, and inserting the result into the inversion equations, Eqs. (29)–(32), leads to the ellipses in Fig. 4b.

As in the case of a univariate signal, there is a distinction between the generating and the inferred, or canonical, ellipse parameters, both of which lead to the same time-varying ellipse. These sets of parameters are identical for a sinusoidally orbited fixed ellipse, and are expected to become increasingly different as the modulation strength increases. Further examination of the conditions for exact recovery of the generating ellipse parameters is outside the scope of this paper.

The generalization of the univariate instantaneous frequency, Eq. (15), to the multivariate case is called the *joint instantaneous frequency*

$$\omega(t) \equiv \frac{\Im\left\{\mathbf{x}^{+H}(t)\mathbf{x}^{+\prime}(t)\right\}}{\|\mathbf{x}^+(t)\|^2} \tag{35}$$





as introduced by Lilly and Olhede (2010b). For the case of a bivariate signal this can be rewritten as

$$\omega(t) = \frac{a_x^2(t)\omega_x(t) + a_y^2(t)\omega_y(t)}{a_x^2(t) + a_y^2(t)} \tag{36}$$

which is seen to be the power-weighted average of the $x$- and $y$-component instantaneous frequencies. Note that this definition of $\omega(t)$ implies that it will generally be positive, and that its power-weighted integral is guaranteed to be non-negative. This is

most apparent from its frequency-domain form, see Eqs. (48) and (50) in Lilly and Olhede (2010b). Note that the instantaneous frequency will always be written with its time argument, $\omega(t)$, to distinguish it from the Fourier frequency $\omega$ as in $e^{i\omega t}$.

The bivariate instantaneous frequency can be written in terms of the ellipse parameters as (Lilly and Olhede, 2010b)

$$\omega(t) = \phi'(t) + \varsigma(t)\,\theta'(t) \tag{37}$$

and thus involves contributions from the rates of change of both the phase $\phi(t)$ and the orientation angle $\theta(t)$. Variations in

$\phi(t)$ correspond to a particle orbiting a fixed ellipse, while variations in $\theta(t)$ capture the precession of the ellipse itself.

## 3 Ellipse extraction

In the previous section, it was shown that given a trajectory consisting of only a single modulated elliptical signal, a unique assignment of time-varying ellipse parameters that could have generated it may be found by forming the associated analytic signal. Real-world eddies, however, do not occur in isolation, but rather are superposed onto flows due to large-scale turbulence,

other eddies, waves, and any self-propagation tendency of the eddy itself. To adapt the analytic signal method to handle realistic trajectories, we therefore need to incorporate a filtering step. This is accomplished using the continuous wavelet transform, as described next. We remind the reader that the notation list in Table A1 is available as a resource.

### 3.1 A model for Lagrangian trajectories

A Lagrangian instrument records a time-varying longitude $\Theta(t)$ and latitude $\Phi(t)$, both taken to be in radians. For small

deviations about some fixed point $(\Theta_o, \Phi_o)$, the trajectory can be locally approximated as

$$\mathbf{x}(t) \equiv \begin{bmatrix} x(t) \\ y(t) \end{bmatrix} = \begin{bmatrix} (\Theta - \Theta_o)\,R_\oplus \cos\Phi_o \\ (\Phi - \Phi_o)\,R_\oplus \end{bmatrix}, \tag{38}$$

where $x(t)$ and $y(t)$ are the east-west and north-south displacements in a plane tangent to the earth at $(\Theta_o, \Phi_o)$ under the small angle approximation, and $R_\oplus$ is the mean radius of the earth. For trajectories lying within a small domain such as the Gulf of Mexico, the errors on estimated eddy properties resulting from neglecting the full spherical geometry are quite minor. The

Coriolis frequency corresponding to latitude $\Phi(t)$ is $f(t) \equiv 2\Omega \sin\Phi(t)$, where $\Omega = 7.292 \times 10^{-4}$ rad s$^{-1}$ is the Earth's angular rotation rate.

The displacement signal $\mathbf{x}(t)$ is assumed to be composed of two portions

$$\mathbf{x}(t) = \mathbf{x}_\epsilon(t) + \sum_{m=1}^{M_\star} \mathbf{x}_\star^m(t) \tag{39}$$





where the $\mathbf{x}_\star^m(t)$ are modulated elliptical signals of the form given in Eq. (22), and where $\mathbf{x}_\epsilon(t)$ is a background displacement
field that will be described as a stochastic process. The modulated oscillations $\mathbf{x}_\star^m(t)$ capture the displacement of a particle
orbiting the center of a possibly elliptical eddy, as well as the motion of particles within inertial oscillations, while the background $\mathbf{x}_\epsilon(t)$ represents motions due to such processes as geostrophic turbulence and wind-forced flow, as well as translation
due to a possible mean flow or systematic drift, and finally any measurement noise.

The corresponding model for the latitude and longitude signals is

$$
\quad \Phi(t) = \Phi_\epsilon(t) + \sum_{m=1}^{M_\star} \Delta\Phi_\star^m(t) \tag{40}
$$

$$
\Theta(t) = \Im\left\{ \log e^{i\left[ \Theta_\epsilon(t) + \sum_{m=1}^{M_\star} \Delta\Theta_\star^m(t)\right]} \right\}. \tag{41}
$$

Here $\Delta\Phi_\star^m(t)$ and $\Delta\Theta_\star^m(t)$ are respectively latitude and longitude deviations associated with the modulated oscillations,
related to $\mathbf{x}_\star^m(t)$ via a small-angle expansion like the one in in Eq. (38). The combination $\Im\left\{ \log e^{i\vartheta} \right\}$ accounts for the case
in which the longitude crosses the antimeridian at $\pm\pi$. Note that Eq. (40) would require modification for trajectories in the
vicinity of the north pole.

Each of the modulated oscillations $\mathbf{x}_\star^m(t)$ is nonzero only within a continuous interval within the time window over which
$\mathbf{x}(t)$ is defined. The total number of oscillatory components present in a timeseries is denoted by $M_\star$, which may be zero, one,
or more than one. The number of nonzero oscillatory components present at any one time is an important quantity $M(t) \leq M_\star$,
referred to as the *multiplicity*. As with $M_\star$, the multiplicity $M(t)$ may be zero, one, or more than one at each time. Examples in
which the multiplicity $M(t)$ is equal to two are an eddy superposed on an inertial oscillation, or an eddy advected by another
eddy.

The conceptual model of Eq. (39) is an example of an *unobserved components model* (e.g. Nerlove, 1967; Harvey, 1989),
meaning that one believes the observations to be the sum of several components which cannot be observed individually, but
only as part of an aggregate process. We may refer to the $\mathbf{x}_\star^m(t)$ as *latent oscillations*, in the sense of being present but obscured.
The goal of this section is present a method for extracting, as accurately as is possible, oscillatory displacement signals $\mathbf{x}_\star^m(t)$
associated with coherent eddies from an observed trajectory $\mathbf{x}(t)$ in the presence of a background flow $\mathbf{x}_\epsilon(t)$, and to link these
displacements to the physical properties of eddies.

### 3.2 A motivational example

The analytic signal method presented in the previous section is intended for the case in which only a single modulated oscil-
lation, and nothing else, is present in a timeseries. It is not intended to work in the case of a composite signal such as in the
unobserved components model of Eq. (39), and performs poorly in such cases.

To illustrate this, and to motivate the approach developed in this section, we return to synthetic signal shown in Fig. 4a. We
write $\mathbf{x}(t) = \mathbf{x}_\epsilon(t) + \mathbf{x}_\star(t)$ with the synthetic signal chosen to be $\mathbf{x}_\star(t)$, and set the background process $\mathbf{x}_\epsilon(t)$ to consist of
a uniform westward drift at 0.5 cm s$^{-1}$ plus a stochastic component. The latter has a red velocity spectrum with a velocity


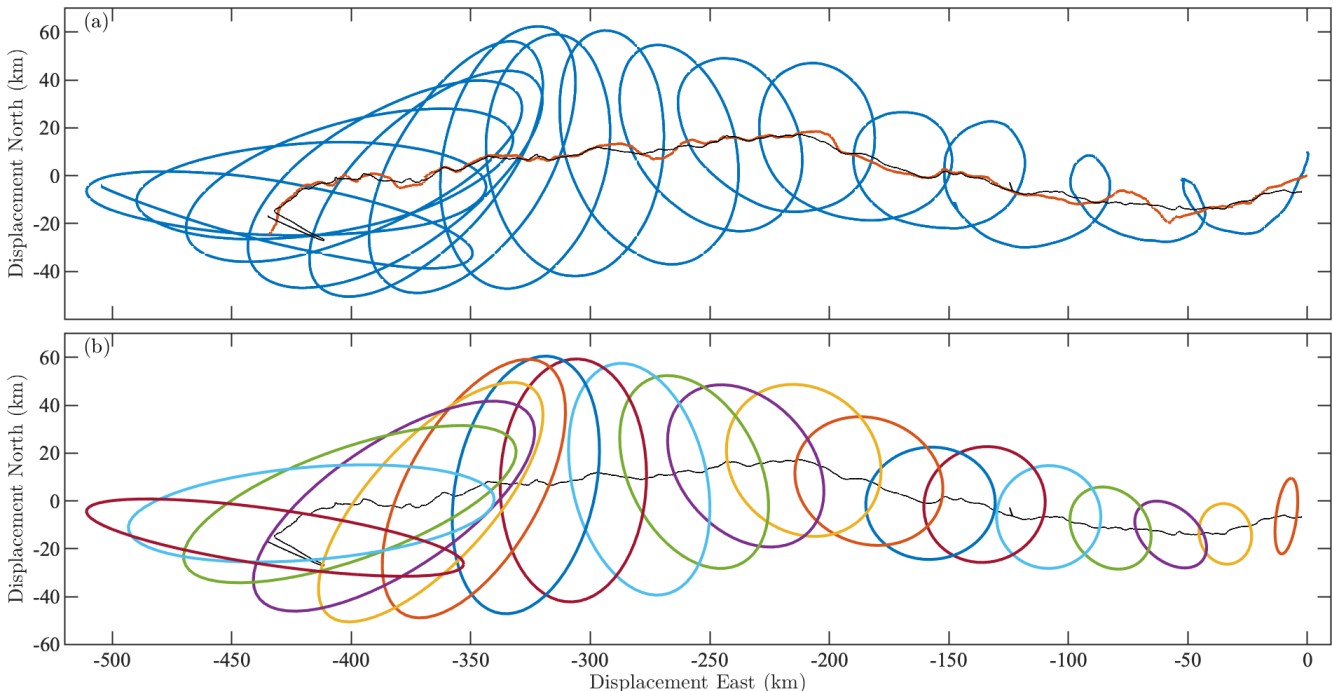

**Figure 5.** Panel (a) shows the signal from Fig. 4a, modified through the addition of a uniform westward drift together with a stochastic component. The sum of the westward drift and the stochastic component is the red curve, which defines the moving center for the modulated elliptical signal. Panel (b) presents ellipses inferred using the wavelet ridge method. The thin black line in both panels is the background $\widehat{\mathbf{x}}_\epsilon(t)$ estimated from the wavelet ridge analysis, an estimate of the red curve in (a).

standard deviation of 0.25 cm s$^{-1}$, constructed as described in Appendix A. The sum of the westward drift and stochastic displacement forms $\mathbf{x}_\epsilon(t)$, shown as the red line in Fig. 5a. The full signal $\mathbf{x}(t)$ is the blue curve in Fig. 5a.

Applying the analytic signal method of the previous section to a detrended version of the total displacement signal $\mathbf{x}(t)$, one obtains the ellipses shown in Fig. 6a. Even though the stochastic signal is small compared to the oscillatory signal, the inferred ellipses are nothing like those we found earlier in Fig. 4b. The reason is that the analytic signal method treats the entire

timeseries as an aggregate, lumping the stochastic signal together with the oscillation. By contrast, the ellipses inferred using the wavelet-based method, developed subsequently and shown in Fig. 6b and also in Fig. 5b, are very close to those seen earlier in Fig. 4b for the monocomponent signal. This is because the wavelet-based method essentially combines the analytic operator with an adaptive filtration, minimizing, though not entirely removing, the influence of the stochastic process.

### 3.3 Wavelet analysis basics

The extraction of unobserved modulated elliptical signals from the observed position signal $\mathbf{x}(t)$ can be accomplished using a method called *multivariate wavelet ridge analysis* (Lilly and Olhede, 2009a, 2012a). This is a generalization to multiple





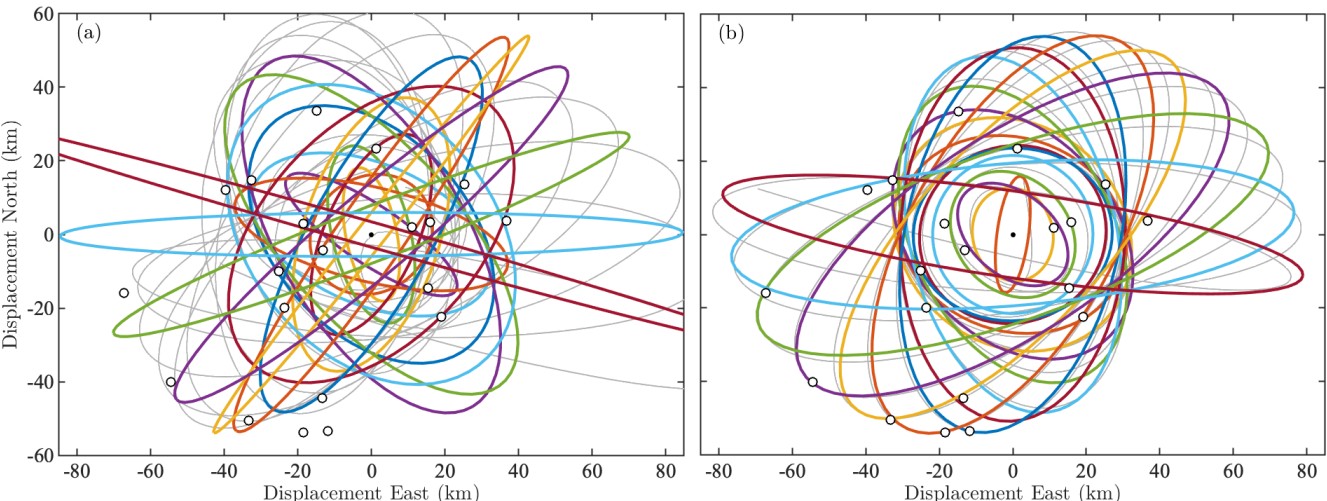

**Figure 6.** Ellipses inferred from the composite signal in Fig. 5a using (a) the analytic signal method directly or (b) the wavelet ridge method formulated in Sect. 3. The gray line in each panel is the estimated modulated elliptical signal using the two respective methods. The dots are as in Fig. 4b.

timeseries of the wavelet ridge analysis method introduced by Delprat et al. (1992), which was extended and refined by Mallat (1999) and later by Lilly and Olhede (2010a). The starting point of wavelet ridge analysis is the choice of a family of oscillatory, time-localized functions, called *wavelets*, that will serve as the basis for detecting modulated oscillations. The choice of a

suitable wavelet function, conventionally denoted $\psi(t)$, involves a tradeoff between time resolution—the ability to resolve sudden changes in an oscillatory signal—and frequency resolution, the ability to distinguish between variability in different frequency bands.

Systematic examination of continuous-time wavelets in common use by Lilly and Olhede (2009b) and Lilly and Olhede (2012b) pointed to the generalized Morse wavelets (Daubechies and Paul, 1988; Olhede and Walden, 2002), defined in the

frequency domain as

$$\Psi_{\beta,\gamma}(\omega) \equiv U(\omega) a_{\beta,\gamma} \omega^{\beta} \mathrm{e}^{-\omega^{\gamma}} \tag{42}$$

as an ideal wavelet family, encompassing all other commonly used forms. Here $U(\omega)$ is again the unit step function, $\beta$ and $\gamma$ are two controlling parameters, and $a_{\beta,\gamma}$ is a normalizing constant defined as

$$a_{\beta,\gamma} \equiv 2(\mathrm{e}\gamma/\beta)^{\beta/\gamma} \tag{43}$$

for reasons to be seen shortly. The time-domain wavelets $\psi_{\beta,\gamma}(t)$ are defined from $\Psi_{\beta,\gamma}(\omega)$ via the inverse Fourier transform,

$$\psi_{\beta,\gamma}(t) \equiv \frac{1}{2\pi} \int\limits_{-\infty}^{\infty} \Psi_{\beta,\gamma}(\omega) \mathrm{e}^{\mathrm{i}\omega t} \mathrm{d}\omega. \tag{44}$$





On account of the unit step function in Eq. (42), these wavelets have vanishing support at negative frequencies and are therefore analytic.

The frequency at which $\Psi_{\beta,\gamma}(\omega)$ obtains its peak magnitude is readily found, from differentiating Eq. (42), to occur at
$\omega_{\beta,\gamma} \equiv (\beta/\gamma)^{1/\gamma}$, called the *peak frequency*. Another key quantity is the normalized curvature of the wavelet at the peak frequency (Lilly and Olhede, 2009b),

$$P_{\beta,\gamma} \equiv \sqrt{-\omega_{\beta,\gamma}^2 \frac{\Psi_{\beta,\gamma}''(\omega_{\beta,\gamma})}{\Psi_{\beta,\gamma}(\omega_{\beta,\gamma})}} = \sqrt{\beta\gamma}. \tag{45}$$

The frequency-domain wavelets can be approximated in the vicinity of their peak frequency as a Gaussian

$$\Psi_{\beta,\gamma}(\omega) = \Psi_{\beta,\gamma}(\omega_{\beta,\gamma}) \, \mathrm{e}^{-\frac{1}{2}P_{\beta,\gamma}^2 \left(\frac{\omega}{\omega_{\beta,\gamma}}-1\right)^2 + f_{\beta,\gamma}(\omega)} \tag{46}$$

as shown in Lilly and Olhede (2012a); here $f_{\beta,\gamma}(\omega)$ is a deviation function, the form of which is given in that reference. We see that $1/P_{\beta,\gamma}$ plays the role of the standard deviation, and is consequently a nondimensional measure of the frequency-domain width of the wavelets about their peak frequency. $P_{\beta,\gamma}$ itself is therefore a nondimensional measure of the wavelet temporal width. Lilly and Olhede (2009b) show that $P_{\beta,\gamma}/\pi$ is roughly the number of oscillations spanned by the central portion time-domain wavelet $|t| \leq P_{\beta,\gamma}/\omega_{\beta,\gamma}$.

With $\gamma$ held fixed, $\beta$ controls the time/frequency tradeoff. Increasing $\beta$ increases the number of oscillations in the wavelet, decreasing the temporal resolution but increasing the frequency resolution. As to the choice of $\gamma$, Lilly and Olhede (2009b) and Lilly and Olhede (2012b) find that the $\gamma = 3$ wavelets are nearly symmetric about their peak in the frequency domain, nearly Gaussian in their frequency-domain shape, and have a standard measure of time/frequency concentration—the Heisenberg area—that is nearly optimal for fixed $\beta$ and any $\gamma$. On account of these desirable properties, the $\gamma = 3$ wavelets are
recommended for general use, and we will follow that recommendation here.

An example of a generalized Morse wavelet is shown in Fig. 7 for the choices $\beta = 2$ and $\gamma = 3$. Here time and frequency have been rescaled, with $\breve{t} \equiv \omega_{\beta,\gamma} t$ and $\breve{\omega} \equiv \omega/\omega_{\beta,\gamma}$, such that the peak frequency occurs at $\breve{\omega} = 1$. This is a quite time-localized wavelet, with at most only two full oscillations visible in the time domain. In the frequency domain, we see that the Gaussian approximation, Eq. (46) with the deviation term $f_{\beta,\gamma}(\omega)$ neglected, is a good approximation to the wavelet and
is indistinguishable from it in a broad region around the peak. The frequencies $\breve{\omega} = 1 \pm 1/P_{\beta,\gamma}$ correspond to one standard deviation of the Gaussian about the peak, values at which the exponential in Eq. (46) is equal to $\mathrm{e}^{-1/2}$. In the time domain, we see slightly less than one full oscillation within the window $|\breve{t}| < P_{\beta,\gamma}$, in agreement with the calculation $P_{\beta,\gamma}/\pi = \sqrt{6}/\pi = 0.8$ as the number of oscillations.

Next we use many rescaled versions of the wavelet to filter a univariate signal $x(t)$, leading to the wavelet transform

$$w_x(t,s) \equiv \int_{-\infty}^{\infty} \frac{1}{s} \psi_{\beta,\gamma}^* \left(\frac{\tau - t}{s}\right) x(\tau) \, \mathrm{d}\tau \tag{47}$$

where the asterisk denotes the complex conjugate, and with $s$ being called the *scale*. The scale $s$ transform amounts to a complex-valued bandpass centered on the rescaled wavelet peak frequency $\omega_{\beta,\gamma}/s$, and thus extracts oscillations having a





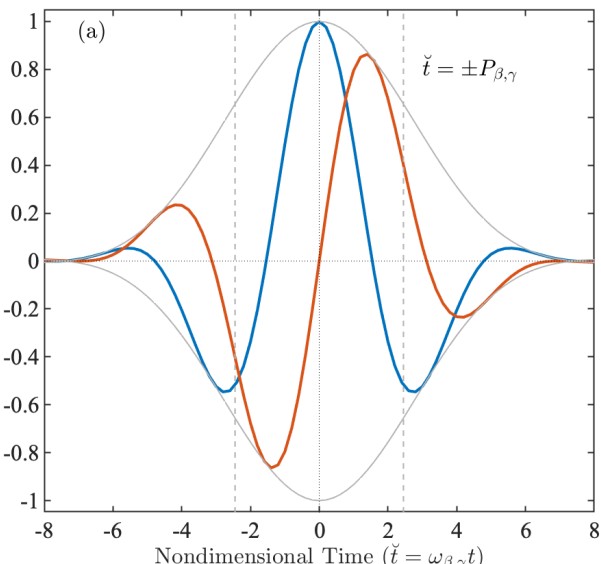
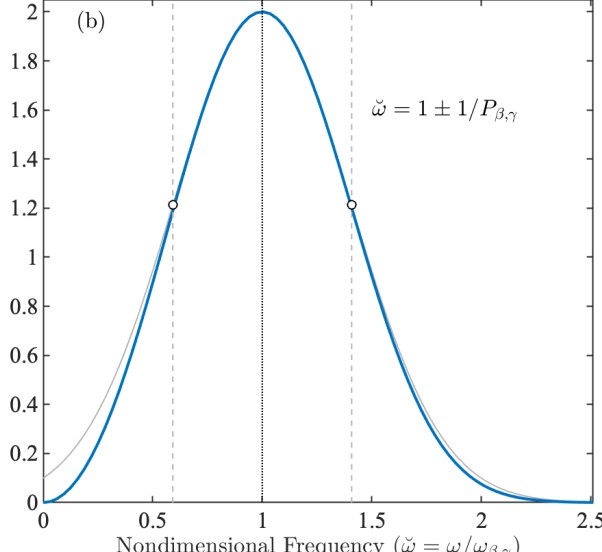

**Figure 7.** A generalized Morse wavelet $\psi_{\beta,\gamma}(t)$ with the choice $(\beta,\gamma) = (2,3)$ in (a) the time domain and (b) its Fourier transform $\Psi_{\beta,\gamma}(\omega)$. Both time and frequency have been nondimensionalized as $\check{t} \equiv \omega_{\beta,\gamma}t$ and $\check{\omega} \equiv \omega/\omega_{\beta,\gamma}$, respectively. In panel (a), the blue and orange lines are the real and imaginary parts, respectively, of $\psi_{\beta,\gamma}(\check{t})$, while the thin gray lines are plus or minus the wavelet modulus $|\psi_{\beta,\gamma}(\check{t})|$. Here the time-domain wavelet has been rescaled to have unit amplitude. In panel (b), the thin gray line is the Gaussian approximation in Eq. (46). Dotted lines mark $\check{t} = 0$ and zero magnitude in (a), and the wavelet peak frequency of $\check{\omega} = 1$ in (b). Vertical dashed lines in both panels are measures of the wavelet half-width based on $P_{\beta,\gamma}$, as marked and as described in the text. Dots in (b) are the wavelet value at $\check{\omega} = 1 \pm 1/P_{\beta,\gamma}$.

period in the vicinity of $2\pi s/\omega_{\beta,\gamma}$. Here we use the $1/s$ scale normalization for the transform, following Lilly and Olhede (2009b). Note that the important dependence of the wavelet transform on the controlling parameters of the wavelet, $\beta$ and $\gamma$, is

suppressed to avoid cumbersome notation. However, it is important to bear in mind that the wavelet transform is not an intrinsic property of a timeseries. Rather, it is a *joint function* of a timeseries and a particular wavelet.

In the frequency domain, the wavelet transform can be expressed as

$$w_x(t,s) = \frac{1}{2\pi} \int_0^\infty \Psi_{\beta,\gamma}(s\omega)X(\omega)\,\mathrm{e}^{\mathrm{i}\omega t}\,d\omega \tag{48}$$

on account of the convolution theorem. This is derived by substituting into Eq. (47) the inverse Fourier transforms of the signal

and the wavelet, Eqs. (16) and (44) respectively[4]. Due to the analyticity of the wavelets, the lower limit of integration may be set to zero. From this one sees that $w_x(t,s)$ consists of a set of versions of $x(t)$ that have been bandpassed using rescaled versions of the wavelet, $\Psi_{\beta,\gamma}(s\omega)$, as the bandpass filters. It is also clear from this expression that the wavelet transform using an analytic wavelet is itself analytic, i.e. supported only on non-negative frequencies, for any fixed scale $s$.

---

[4]The complex conjugate that would normally appear on the wavelet's Fourier transform $\Psi_{\beta,\gamma}(s\omega)$ after applying the convolution theorem has been dropped because these particular wavelets are real-valued in the frequency domain.





With these definitions, the wavelet transform of a phase-shifted sinusoid

$$x_o(t) \equiv a_o \cos(\omega_o t + \phi_o) \qquad (49)$$

is found to be

$$w_o(t,s) = \frac{1}{2} a_o e^{i\omega_o t + i\phi_o} \Psi_{\beta,\gamma}(s\omega_o) \qquad (50)$$

using Eq. (48) together with Eq. (20). We see that $w_o(t,s)$ obtains its maximum magnitude at the scale $s = \omega_{\beta,\gamma}/\omega_o$. Moreover, if one sets the maximum value of the wavelet to be $\Psi_{\beta,\gamma}(\omega_{\beta,\gamma}) = 2$, we have

$$w_o(t, \omega_{\beta,\gamma}/\omega_o) = a_o e^{i\omega_o t + i\phi_o} \qquad (51)$$

such that the real part of the wavelet transform at the maximizing scale $s = \omega_{\beta,\gamma}/\omega_o$ recovers the original signal $x_o(t)$. Owing to the choice of normalizing constant in Eq. (42) as $a_{\beta,\gamma} \equiv 2(e\gamma/\beta)^{\beta/\gamma}$, we have $\Psi_{\beta,\gamma}(\omega_{\beta,\gamma}) = 2$ as desired.

### 3.4  Oscillation detection using wavelet ridge analysis

Wavelet ridge analysis is a method for extracting modulated oscillations from a timeseries using the wavelet transform. Turning
now to a vector-valued signal $\mathbf{x}(t)$, its wavelet transform will be a vector function of time and scale,

$$\mathbf{w}(t,s) \equiv \int_{-\infty}^{\infty} \frac{1}{s} \psi_{\beta,\gamma}^* \left( \frac{\tau - t}{s} \right) \mathbf{x}(\tau) \, \mathrm{d}\tau = \begin{bmatrix} w_x(t,s) \\ w_y(t,s) \end{bmatrix}. \qquad (52)$$

A *ridge* of this vector-valued wavelet transform $\mathbf{w}(t,s)$ is a time-varying scale curve $\hat{s}(t)$ along which

$$\frac{\partial}{\partial s} \|\mathbf{w}(t,s)\| = 0, \qquad \frac{\partial^2}{\partial s^2} \|\mathbf{w}(t,s)\| < 0 \qquad (53)$$

such that $\hat{s}(t)$ marks a local maximum of the wavelet transform magnitude with respect to variations in scale $s$. This is the
multivariate ridge definition introduced by Lilly and Olhede (2009a) and Lilly and Olhede (2012a), building on earlier work on univariate wavelet ridge analysis by Delprat et al. (1992), Mallat (1999), and Lilly and Olhede (2010a). It was shown by Lilly (2017) that a ridge can be interpreted as a maximum with respect to scale of the local signal energy density, or power, see Section 2(c) therein. The principle of wavelet ridge analysis is therefore not one of minimizing an error, but rather of maximizing the power of a projection.[5]

For a vector-valued signal with only a single modulated oscillation present, $\mathbf{x}(t) = \mathbf{x}_\star(t)$, it can be demonstrated that a wavelet ridge approximates the location of the instantaneous frequency curve $\omega(t)$, that is,

$$\hat{s}(t) \approx \frac{\omega_{\beta,\gamma}}{\omega(t)} \qquad (54)$$

---

[5]The first author thanks G. Sutyrin for asking the question, "What is the principle?".





while the estimated analytic signal associated with $\mathbf{x}(t)$ is found by simply taking the value of the wavelet transform along the ridge,

$$\widehat{\mathbf{x}}^+(t) \equiv \mathbf{w}\left(t, \hat{s}(t)\right) \approx \mathbf{x}^+(t). \tag{55}$$

These equations are derived in Lilly and Olhede (2012a), their Eq. (86) and Eq. (61) respectively, wherein the explicit forms for next-order terms in the approximations are given. Those terms, omitted for brevity here, represent error in the wavelet ridge method arising from the strength of the modulated oscillation itself and therefore are a type of bias error. Such errors vanish when the modulation strength vanishes, that is, for the case of a sinusoidal signal.

When applied to a real-world signal $\mathbf{x}(t)$, wavelet ridge analysis leads to a set of ridges, $\hat{s}^m(t)$, the corresponding estimated oscillations $\widehat{\mathbf{x}}_\star^m(t)$, their associated ellipse parameters $\widehat{\kappa}^m(t)$, $\widehat{\varsigma}^m(t)$, $\widehat{\theta}^m(t)$, and $\widehat{\phi}^m(t)$, and finally the instantaneous frequencies $\widehat{\omega}^m(t)$, for $m = 1, 2 \ldots \widehat{M}_\star$. Here $\widehat{M}_\star$ is the total number of ridges present after applying any editing criteria, such as that discussed in the next section. These estimates are interpreted here in the context of the unobserved components model, Eq. (39). We would say that the ridges provide estimates of latent oscillations that appear to be present in the timeseries, with $\widehat{M}_\star$ being an estimate of the total number $M_\star$ of such components. The dependence of all of these estimates on the wavelet properties $(\beta, \gamma)$ is implicit. For notational convenience, we will drop the superscripts "$m$" on these quantities, letting it be understood that they are estimates pertaining to a particular latent oscillation.

Application of the ridge analysis method to surface drifter trajectories leads to another type of error, namely 'false positives' arising from the stochastic background $\mathbf{x}_\epsilon(t)$. As discussed in the Introduction, the ridge analysis will identify some events that, rather than corresponding to latent oscillations that are actually present, arise by chance due to the interaction of the wavelets with the background. These are a major problem because they mean a certain fraction of detected ridges would be expected to be entirely spurious. A means of addressing the false positives is presented in Sect. 4.

If the multiplicity exceeds one, we may define the time-varying position of the center of the $m$th oscillation as

$$\mathbf{x}_\diamond^m(t) \equiv \mathbf{x}(t) - \mathbf{x}_\star^m(t) = \mathbf{x}_\epsilon(t) + \sum_{n=1}^{M_\star, n \neq m} \mathbf{x}_\star^n \tag{56}$$

which is the difference between the full displacement signal and the displacement signal associated only with that ridge. This quantity is estimated by as $\widehat{\mathbf{x}}_\diamond^m(t) \equiv \mathbf{x}(t) - \widehat{\mathbf{x}}_\star^m(t)$. To put this in terms of latitude and longitude, we first convert $\widehat{x}_\star^m(t)$ and $\widehat{y}_\star^m(t)$ into the estimated deviations

$$\Delta\widehat{\Theta}_\star^m(t) \equiv \frac{\widehat{x}_\star^m(t)}{R_\oplus \cos\Phi_o}, \qquad \Delta\widehat{\Phi}_\star^m(t) \equiv \frac{\widehat{y}_\star^m(t)}{R_\oplus} \tag{57}$$

where $\Phi_o$ is latitude of the reference point that was used to create the Cartesian vector $\mathbf{x}(t)$ from latitude and longitude in Eq. (38). Then

$$\widehat{\Phi}_\diamond^m(t) \equiv \Phi(t) - \Delta\widehat{\Phi}_\star^m(t) \tag{58}$$

$$\widehat{\Theta}_\diamond^m(t) \equiv \Im\left\{\log e^{i\left[\Theta(t) - \Delta\widehat{\Theta}_\star^m(t)\right]}\right\} \tag{59}$$





gives the estimated latitude and longitude position of the center of the $m$th oscillation.

Finally, the background process on the Cartesian plane, $\mathbf{x}_\epsilon(t)$, is estimated by summing over all ridges and forming the
residual

$$\widehat{\mathbf{x}}_\epsilon(t) \equiv \mathbf{x}(t) - \sum_{m=1}^{\widehat{M}_\star} \widehat{\mathbf{x}}_\star^m(t). \tag{60}$$

The corresponding latitude and longitude signals, $\widehat{\Phi}_\epsilon(t)$ and $\widehat{\Theta}_\epsilon(t)$, are formed by subtraction after inserting the estimated deviations $\Delta\widehat{\Phi}_\star^m(t)$ and $\Delta\widehat{\Theta}_\star^m(t)$ into Eqs. (40) and (41).

### 3.5  A ridge length threshold

Experience shows that it is important to set a threshold for the minimum length of a ridge. The ridge length is measured in terms of the number of oscillations executed along the ridge, which is found by integrating the estimated instantaneous frequency $\widehat{\omega}(t)$

$$L_\star \equiv \frac{1}{2\pi} \int_{t_i}^{t_f} \widehat{\omega}(t)\mathrm{d}t \tag{61}$$

from the initial time point of the ridge $t_i$ to its final time point $t_f$. Note that $L_\star$ is nondimensional. For a constant instantaneous
frequency, $\hat{\omega}(t) = \omega_o$, observed over a time interval of duration $T = t_f - t_i$, we have $L_\star = T/(2\pi/\omega_o)$ which is the number of periods executed over time interval $T$.[6]

A threshold on $L_\star$ is found to be important for two reasons. Firstly, when any kind of noise is present, the random generation of false positives becomes more common as the ridge length decreases; thus $L_\star$ is important for assessing statistical significance, as will be seen in more detail shortly. Secondly, as the ridge length becomes small compared to the number of
oscillations experienced by the wavelet, the very idea that the wavelet transform is illuminating some aspect of the timeseries becomes suspect. For ridges comparable to or shorter than the length of the wavelet, isolated noise points or minor discontinuities in the timeseries lead, via the convolution theorem, to copies of the wavelet itself being presented in the wavelet transform. As mentioned earlier the wavelet transform is a joint function of the wavelet and the signal; ridges comparable to or shorter than the wavelet duration tend to be features arising from the former.

A ridge length threshold is therefore employed in which we keep only ridges with lengths

$$L_\star > n \frac{2P_{\beta,\gamma}}{\pi} \tag{63}$$

---

[6]In the case of a univariate signal $x(t)$, we have

$$L_\star \equiv \frac{1}{2\pi} \int_{t_i}^{t_f} \widehat{\omega}_x(t)\mathrm{d}t = \frac{1}{2\pi}\left[\widehat{\phi}_x(t_f) - \widehat{\phi}_x(t_i)\right] \tag{62}$$

and $L_\star$ can be interpreted in terms of the phase change along the ridge. For the multivariate case, the multivariate instantaneous frequency of Eq. (36) is no longer defined as the derivative of some phase, so it would appear this interpretation no longer holds.





for some choice of $n$, where $2P_{\beta,\gamma}/\pi$ is approximately the total number of oscillations spanned by the wavelet. Initially we choose $n = 1$ so that $L_\star > 2\sqrt{6}/\pi \approx 1.6$, where as mentioned earlier we take $\beta = 2$ and $\gamma = 3$. A more refined value of the cutoff will be determined later as a part of determining which ridges are statistically significant.

### 3.6  A synthetic example

As a simple example, the wavelet ridge analysis for the synthetic composite signal from Fig. 5 is shown in Fig. 8, using the $\psi_{2,3}(t)$ wavelet from Fig. 7 and the ridge length criterion $L_\star > 2\sqrt{6}/\pi$. The white and black lines denote ridge curves $\hat{s}(t)$. A clear maximum of the wavelet transform magnitude $\|\mathbf{w}(t,s)\|$ with respect to scale is observed, marked by the heavy black line. The white lines are short ridges marking minor, generally indiscernible maxima of the wavelet transform. These are spurious features arising randomly due to the stochastic background process $\mathbf{x}_\epsilon(t)$, and are rejected based on the ridge length criterion. After this rejection, only the black curve remains. Thus the wavelet ridge analysis supports the unobserved components model with multiplicity one, $\mathbf{x}(t) = \mathbf{x}_\epsilon(t) + \mathbf{x}_\star(t)$, and we interpret the remaining ridge as an estimate of the latent oscillatory signal $\mathbf{x}_\star(t)$.

Evaluating the wavelet transform along the black ridge curve $\hat{s}(t)$ as in Eq. (55) leads to an estimate of the analytic signal vector $\widehat{\mathbf{x}}_\star^+(t)$ associated with the latent oscillation $\mathbf{x}_\star(t)$. Application of the ellipse inversion equations, Eqs. (29)–(32), gives the estimated ellipses shown in Fig. 5b and Fig. 6b. Taking the real part of this estimated analytic signal produces an estimate of $\mathbf{x}_\star(t)$ itself, $\widehat{\mathbf{x}}_\star(t) = \Re\{\widehat{\mathbf{x}}_\star^+(t)\}$, the gray line in Fig. 6b, with its corresponding estimated velocities shown in Fig. 8a as gray lines. Subtracting the estimated oscillation from the full signal leads to the estimated background, $\widehat{\mathbf{x}}_\epsilon(t) \equiv \mathbf{x}(t) - \widehat{\mathbf{x}}_\star(t)$, the black line in Figs. 5a,b.

It is clear from Figs. 4–6 that the wavelet ridge analysis does an excellent job of extracting the true oscillation $\mathbf{x}_\star(t)$ from the composite signal $\mathbf{x}(t)$, effectively stripping out the background process $\mathbf{x}_\epsilon(t)$. Note that this signal, like realistic eddy signals, exhibits substantial frequency modulation, and thus Fourier-based or narrowband methods such as bandpassing or complex demodulation would not be suitable.

Importantly, we have not needed to specify any properties of the oscillatory signal apart from the frequency range of the transform. As mentioned earlier, Lilly and Olhede (2009b) and Lilly and Olhede (2012b) have established the sensibility of the choice $\gamma = 3$ based on symmetry considerations, thus the only free parameter of the wavelet transform is $\beta$, which controls the number of oscillations in the wavelet. The choice of $\beta$, it should mentioned, encodes an implicit tradeoff between resolving temporal variability of detected modulated oscillations, and separating an oscillation from neighboring variability in the frequency domain. Smaller values of $\beta$ are more suitable for strong modulation, whereas large values would be more appropriate if it were necessary to resolve closely spaced oscillations in frequency. The $\beta = 2$ value used here, corresponding to quite narrow wavelets in the time domain, appears to be suitable for this and many other Lagrangian datasets.

### 3.7  One-sided ridges

Because here we are particularly interested in eddies, which tend to be destroyed if strained to a high degree of eccentricity, a modification is made to the ridge analysis. In its general form, the ridge analysis places no constraints on the polarization, that


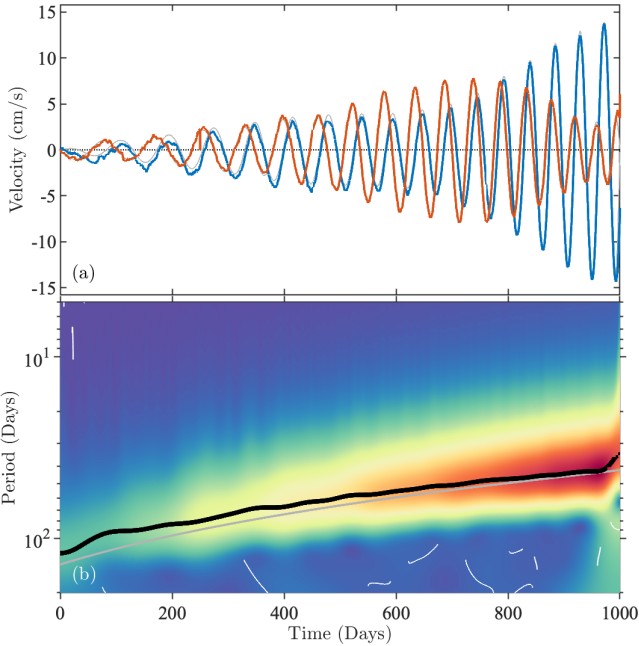

**Figure 8.** Wavelet ridge analysis applied to the synthetic composite signal shown in Fig. 5a. Panel (a) is the velocity corresponding to the displacement signal, with blue for the eastward component and orange for the northward component. The magnitude of the wavelet transform of the displacement signal $\|\mathbf{w}(t,s)\|$ using a $(\beta,\gamma) = (2,3)$ generalized Morse wavelet is shown in (b), with the heavy black curve denoting a major ridge and the white curves denoting short, spurious ridges arising as a result of the stochastic background process. The period corresponding to the generating parameters is shown as the gray line, see Appendix A. Thin gray lines in panel (a) are the velocity components formed by differentiating the ridge-based estimate of the oscillatory displacement signal $\widehat{\mathbf{x}}_{\star}(t)$ formed from the real part of Eq. (55).

is, the degree of eccentricity of the signal. Numerical experiments with noise datasets show that when ridges emerge, they can be of any polarization, and this polarization tends to wander with time, sometimes changing sign across $\varsigma(t) = 0$ as the ridge switches from being dominated by positive rotations to dominated by negative rotations. Such transitions are quite unrealistic for real-world eddies, and permitting them in the ridge analysis leads to a substantial number of false positives which must be sorted out later.

It therefore seems preferable to explicitly exclude sign transitions across $\varsigma(t) = 0$. Such ridges will be said to be *one-sided*, and are formed as follows. The Cartesian wavelet transforms are converted into a pair of rotary transforms as

$$\begin{bmatrix} w^+(t,s) \\ w^-(t,s) \end{bmatrix} = \frac{1}{\sqrt{2}} \begin{bmatrix} 1 & i \\ 1 & -i \end{bmatrix} \mathbf{w}(t,s) \tag{64}$$

with $|w^+(t,s)|$ giving the local magnitude of positive oscillations in $z(t) = x(t) + iy(t)$, and $|w^-(t,s)|$ giving the magnitude of negative oscillations, see e.g. Lilly and Gascard (2006). As this matrix transformation is unitary, we have $\|\mathbf{w}(t,s)\| = $





$\sqrt{|w^+(t,s)|^2 + |w^-(t,s)|^2}$. We then form $\Delta w(t,s) \equiv |w^+(t,s)| - |w^-(t,s)|$, the difference of the positive and negative rotary wavelet transform amplitudes.

The wavelet ridge analysis is then performed twice. Ridges of $\mathbf{w}(t,s)$ are first found after masking out regions where $\Delta w(t,s) > 0$, and then again after masking out regions where $\Delta w(t,s) < 0$. Each of these two sets of ridges lacks sign transitions. Before applying the ridge length threshold, Eq. (63), the union of these two sets of ridges contains the same ridge

points as the full set of ridges, with the exception of any ridge points for which $\Delta w(t,s)$ is exactly zero. After applying the ridge length cutoff, the set based on the one-sided ridges will be smaller than the full set, because sign transitions are essentially treated as ridge endpoints.

The net result of this modification is that ridges lacking a sign transition are unchanged, ridges containing sign transitions are broken into shorter segments, and far fewer spurious, false positive events survive the ridge length threshold. A desirable

feature of this approach is that it does not involve setting an *ad hoc* threshold on the degree of eccentricity, as any numerical value of the circularity $\varsigma(t)$ may still be detected provided the rotation sense does not reverse.

## 4 Application to the Gulf of Mexico drifter dataset

In this section the wavelet ridge analysis method is applied to the Gulf of Mexico drifter dataset presented in Fig. 1. Issues of false positives are addressed, leading to an edited eddy dataset where such features are believed to be rare. Note that in this

section we omit the "$m$" superscripts on modulated oscillation and ridge quantities to avoid excessive notation.

### 4.1 Choice of frequency band

The first decision to be made is what band of scales, or frequencies, the wavelet transform should be performed over. The instantaneous frequency $\omega(t)$ can be nondimensionalized relative to the Coriolis frequency at the center of the oscillation, $f_\diamond(t) \equiv 2\Omega \sin \widehat{\Phi}_\diamond(t)$, as

$$\widetilde{\omega}(t) \equiv \mathrm{sgn}\left(\varsigma(t)\right) \frac{\omega(t)}{f_\diamond(t)} \tag{65}$$

which we construct to be a signed quantity, positive for cyclonic motion and negative for anticyclonic motion. We choose to carry out the ridge analysis within a time-varying frequency band such that the nondimensional instantaneous frequency is in the broad range $1/64 \leq |\widetilde{\omega}(t)| \leq 2$. This range is expected to capture eddy-related motions from slow circling on the far flanks of large eddies, up to the rapid oscillations within highly nonlinear submesoscale eddies.

It is helpful to examine the physical interpretation of the nondimensional instantaneous frequency $\widetilde{\omega}(t)$ by referring to the special case of a steady circular eddy with azimuthal velocity profile $v_\theta(r)$. This eddy has vertical vorticity $\zeta(r)$ and angular velocity $\varpi(r)$ given by

$$\zeta(r) \equiv \hat{\mathbf{k}} \cdot \nabla \times \mathbf{v} = \frac{v_\theta(r)}{r} + \frac{\partial}{\partial r} v_\theta(r) \tag{66}$$

$$\varpi(r) \equiv \hat{\mathbf{k}} \cdot \frac{\mathbf{r} \times \mathbf{v}}{\|\mathbf{r}\|^2} = \frac{v_\theta(r)}{r} \tag{67}$$





with $\hat{\mathbf{k}}$ being the vertical unit vector. It follows from the latter that $z_o(t) = r_o e^{i\varpi(r_o)t}$ are solutions for particle paths about the eddy center at some fixed radius $r_o$. Thus for the case of constant angular velocity in a circular eddy, the angular velocity is the same as the signed instantaneous frequency, $\varpi(r_o) = \operatorname{sgn}(\varsigma(t))\,\omega(t)$.

Let the eddy have a maximum azimuthal velocity at radius $R_o$ with a value of $V_o = v_\theta(R_o)$. Its nonlinearity may be measured through the vortex velocity Rossby number,

$$Ro_V(t) \equiv \frac{2V_o}{R_o f(t)} \tag{68}$$

see e.g. D'Asaro et al. (1994). (Note that the function of time arises because in this simple kinematic model we permit—unrealistically—the fixed eddy to move meridionally without adjustment.) One may extend the vortex Rossby number to measure the local nonlinearity of the flow at each radius via

$$Ro_V(r,t) \equiv \frac{2v_\theta(r)}{r f(t)} = \operatorname{sgn}(\varsigma(t))\frac{2\omega(t)}{f(t)} = 2\widetilde{\omega}(t). \tag{69}$$

The equalities on the right-hand side show that for a steady circular eddy, the local Rossby number $Ro_V(r,t)$ is twice the nondimensional instantaneous frequency $\widetilde{\omega}(t)$ that would be observed for a particle located at radius $r$.

Another measure of the bulk nonlinearity of an eddy is its vorticity Rossby number, defined as

$$Ro_\varsigma(t) \equiv \frac{\zeta_o}{f(t)} \tag{70}$$

with $\zeta_o$ being the vertical vorticity averaged over the radius $R_o$. In the case that the eddy core is in solid-body rotation, then
$v_\theta(r) = \alpha r$ for some constant $\alpha$ with $r \le R_o$, and the core vorticity is $\zeta(r) = 2v_\theta(r)/r = 2\alpha$ from Eq. (66). It follows that $Ro_\varsigma$ and $Ro_V$ are identical for solid-body rotation.

If a time-varying lower boundary is set for the ridge frequency band at $\widetilde{\omega}(t) = -1/2$, it is believed on dynamical grounds that that no anticyclonic eddies will be missed, and also that inertial oscillations will be entirely excluded, as we now show. It is well known that for anticyclonic, circular, barotropic eddies, there is a stability boundary at $\zeta/f = -1$; see Kloosterziel
(2010) for a very useful historical discussion of the relevant stability criterion. Eddies having more negative values of relative vorticity would experience a type of instability known as inertial or centrifugal instability. Consequently such eddies are not observed in nature, with the most intense documented anticyclones, such as the Lofoten Basin Eddy (Søiland and Rossby, 2013; Bosse et al., 2019; Trodhal et al., 2020), having vorticities of around $\zeta = -f$. This stability boundary corresponds to a nondimensional frequency of $\widetilde{\omega}(t) = -1/2$.

Inertial oscillations typically occur at a Fourier frequency of $\omega = -f$ or $\widetilde{\omega}(t) = -1$, with the negative sign indicating that inertial oscillations turn clockwise in the Northern Hemisphere and counterclockwise in the Southern Hemisphere. However, in the presence of background vorticity, inertial oscillations experience a frequency shift (e.g. Kunze, 1985) from $\omega = -f$ to $\omega = -f - \zeta/2$. For the most extreme possible anticyclonic eddy at the stability boundary $\zeta/f = -1$, assuming solid-body rotation, the shifted inertial frequency would then occur at $\omega = -f/2$ or $\widetilde{\omega}(t) = -1/2$, which is the same as frequency of the
eddy itself. Thus inertial shifting in an anticyclone brings the inertial oscillation frequency away from $f$ and closer to the eddy frequency, and these become identical for the most nonlinear anticyclones at $\zeta/f = -1$.





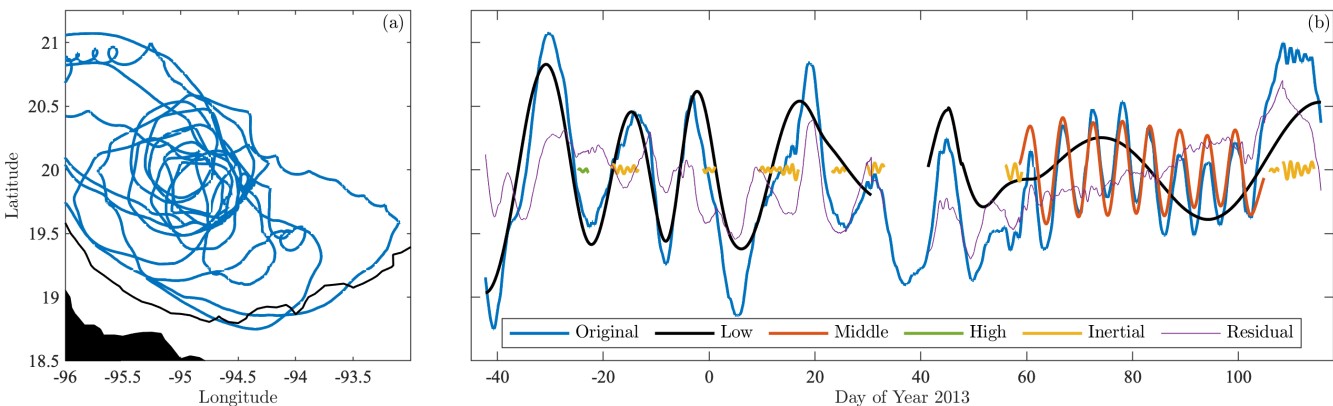

**Figure 9.** An example of a trajectory decomposed according to the unobserved components model of Eq. (39), using the one-sided wavelet ridge method. Panel (a) shows the trajectory of drifter ID #80348 from the SGOM experiment, in the Campeche Gyre in the southwestern Gulf of Mexico. In panel (b), the original latitude signal of this trajectory is shown, along with oscillatory signals extracted from low, medium, and high frequencies, as well as the inertial band. Also shown is the residual after adding up all the extracted signals and subtracting the result from the original, Eq. (60), converted back to a latitude. The origin of the various curves is shown in Fig. 10.

Therefore, both to avoid inertial oscillations and because no eddies are physically expected, the ridge analysis should be truncated to exclude events below the time-varying curve $\widetilde{\omega}(t) = -1/2$. Yet since there is no corresponding stability boundary on the cyclonic side, one should anticipate the possibility of intense cyclonic eddies exceeding $\widetilde{\omega}(t) = 1/2$. The truncation to

$\widetilde{\omega}(t) = -1/2$ will therefore be done at a later stage once we scrutinize the initial results using the symmetric band $1/64 \le |\widetilde{\omega}(t)| \le 2$.

### 4.2   An example from the Bay of Campeche

An example of applying the one-sided ridge analysis to a drifter trajectory from the Bay of Campeche in the southwestern Gulf of Mexico is presented in Figs. 9 and 10. The trajectory is shown in Fig. 9a, its latitude timeseries is the blue line in Fig. 9b, and

its zonal and meridional velocities are presented in Fig. 10a. The trajectory has a complex oscillatory behavior, executing orbits that appear significantly non-circular, sometimes even square, and with numerous small cusps or loops. The latitude signal and velocity timeseries both reveal substantial nonstationary, with a transition from low-frequency variability to higher-frequency variability around yearday 50 of 2013, and to still higher-frequency variability around yearday 110. The roughness observed in both position and velocity is suggestive of superposed variability from smaller temporal scales.

This trajectory is analyzed using the same $(\beta, \gamma) = (2, 3)$ generalized Morse wavelet employed previously. The range of scales in the transform is chosen to encompass the nondimensional frequency range $1/64 \le |\widetilde{\omega}(t)| \le 2$ for all latitudes within the Gulf. The total transform magnitude $\|\mathbf{w}(t, s)\|$, positive rotary transform magnitude $|w^+(t, s)|$, and negative rotary transform magnitude $|w^-(t, s)|$ are shown in Fig. 10b–d. The colored lines in Fig. 10b–d are all the one-sided ridges—that is, scale curves $\hat{s}(t)$ satisfying Eq. (53), yet lacking sign transitions of $\hat{\varsigma}(t)$—subject to the minimum ridge length of $L_\star = 2\sqrt{6}/\pi \approx 1.6$.



**Figure 10.** One-sided wavelet ridge analysis of the signal presented in Fig. 9. In (a), the zonal and meridional velocities $u(t)$ and $v(t)$ associated with the displacement signal in Fig. 9a are shown as blue and orange curves, respectively. Panel (b) shows the total wavelet transform magnitude, $\|\mathbf{w}(t,s)\| = \sqrt{|w^+(t,s)|^2 + |w^-(t,s)|^2}$, while (c) and (d) show the magnitudes of the positive and negative rotary transforms, $|w^+(t,s)|$ and $|w^-(t,s)|$. The y-axis in all of these plots is scale converted to oscillation period, $2\pi s/\omega_{\beta,\gamma}$. The thin, nearly horizontal gray line in panels (b)–(d) is the scale location of the time-varying Coriolis frequency $f(t)$, while the thin white lines mark the curves where the magnitudes of the two rotary transforms are equal, $|w^+(t,s)| = |w^-(t,s)|$. The colored lines show all detected ridges $\widehat{s}(t)$ of $\|\mathbf{w}(t,s)\|$ using the same color scheme as in Fig. 9b. There are two sets of ridges, those detected in $\|\mathbf{w}(t,s)\|$ when the positive rotary transform $w^+(t,s)$ dominates, and those detected in $\|\mathbf{w}(t,s)\|$ when the negative rotary transform $w^+(t,s)$ dominates. To show their different origins, the ridges are superposed on the dominant transform as well as being drawn on the total transform $\|\mathbf{w}(t,s)\|$. A condition of the one-sided algorithm is that the ridges terminate when they encounter the thin white line, prohibiting transitions in rotation sense.





Each ridge is drawn on the rotary transform that is dominant over its duration, as well as on the total transform in Fig. 10b. The one-sided ridges are not permitted to cross the white lines, which mark the locations at which $|w^+(t,s)| = |w^-(t,s)|$.

     The nonstationarity and multi-scale variability that are apparent by eye are seen explicitly in the wavelet transforms. Small-scale variability is frequently attributable to inertial oscillations, as well as to a brief high-frequency cyclonic event around yearday -22. At lower frequencies, a bifurcation is seen around yearday 58, where a low-frequency cyclonic ridge splits into

two cyclonic ridges, one that descends to lower frequencies and one that rises to higher frequencies. Whereas the higher-frequency member of this pair is visible by eye in the velocity and position timeseries, the lower-frequency member is not. The two low-frequency ridges that together span most of the record are notable for having relatively high values of eccentricity, unlike the middle-frequency ridge in the yearday range 58–105, which is seen to be strongly circularly polarized in the cyclonic sense.

The contributions of the various ridges to the latitude variability is seen in Fig. 9b, after converting the ridges to latitude $\widehat{\Phi}^m_\star(t)$ deviations using Eq. (57), as is the latitude residual $\widehat{\Phi}_\epsilon(t)$. Oscillatory variability is much reduced compared with the original, particularly during the latter portion of the record dominated by the middle-frequency ridge. An exception is in the middle of the record, in the vicinity of the gap between the two low-frequency ridges, where the dominant wavelet polarization is observed to briefly change sign for a reason that is not readily apparent.

This complex trajectory is a good example of a situation in which the multiplicity $\widehat{M}(t)$—the apparent number of modulated oscillations at any moment—is greater than one. The presence of inertial oscillations superposed on a background mesoscale eddy, which accounts for the cusps seen in Fig. 9a, is fairly common in this dataset. The superposition of two lower-frequency ridges occurs less frequently. A physical hypothesis consistent with two contemporaneous low-frequency ridges is that a particle is trapped in an eddy that is itself being advected by another eddy, with the lower-frequency signal arising from advection

on the exterior flank of the second eddy. This appears to be the situation during yeardays 60–110. The superposition of these two signals accounts for the "wobble" in Fig. 9a, where the center of the tight loops in the middle of the plot appears to vary over time.

### 4.3   A noise dataset for a null hypothesis

Because the eddies we are interested in studying do not occur in isolation, but are embedded within a turbulent background

flow, it is necessary to take into account that the background flow itself may occasionally, by chance, give rise to features in the drifter trajectories that appear as modulated oscillations. Such features would be detected by the ridge analysis and therefore constitute false positives.

     To address this issue, we will compare the results of the ridge analysis to the results of a parallel analysis applied to a stochastic or "noise" dataset. The noise dataset will be created to match key properties of the observed dataset, but lacking the

explicit signatures of any eddies. This will act as a null hypothesis and enable a level of statistical significance to be determined. It amounts to a idealized approximation to the background process $\mathbf{x}_\epsilon(t)$ from the unobserved components model of Eq. (39) that is constrained to be isotropic, in other words, lacking a preference for fluctuations in the zonal vs. meridional direction as well as in the positive vs. negative rotational senses.




The noise dataset is constructed as follows. For each trajectory, we form an estimate of the spectrum of the complex-valued
velocity $\nu(t) \equiv u(t) + \mathrm{i}v(t) = z'(t)$, denoted $S_{\nu\nu}(\omega)$. It is known that $S_{\nu\nu}(\omega)$ gives contributions to the velocity variance
from positively-rotating Fourier components for $\omega > 0$, and from negative-rotating Fourier components for $\omega < 0$, see Gonella
(1972). Thus $S_{\nu\nu}(\omega)$ is said to be the *rotary spectrum* associated with $\nu(t)$. We use the multitaper method (Thomson, 1982;
Park et al., 1987; Percival and Walden, 1993) to create an estimated spectrum $\widehat{S}_{\nu\nu}(\omega)$. The implicit degree of frequency-domain
smoothing is specified by choosing the taper time-bandwidth product equal to four, a setting that leads to relatively smooth
spectral estimates for this data. Prior to forming the spectra, the temporal means of the velocity series are removed in order to
prevent leakage from the zero-frequency component, a common processing step that has the effect of setting $\widehat{S}_{\nu\nu}(0) = 0$.

At each frequency, we define the spectrum of the complex-valued noise velocities, which will be denoted $\varepsilon(t)$, to be the
minimum of the two sides of the rotary spectrum

$$S_{\varepsilon\varepsilon}(\omega) \equiv c_\varepsilon \min\left\{\widehat{S}_{\nu\nu}(\omega), \widehat{S}_{\nu\nu}(-\omega)\right\} \tag{71}$$

where $c_\varepsilon$ is a normalization factor given by

$$c_\varepsilon \equiv \frac{\frac{1}{2\pi}\int_{-\infty}^{\infty} \widehat{S}_{\nu\nu}(\omega)\mathrm{d}\omega}{\frac{1}{2\pi}\int_{-\infty}^{\infty} \min\left\{\widehat{S}_{\nu\nu}(\omega), \widehat{S}_{\nu\nu}(-\omega)\right\}\mathrm{d}\omega}. \tag{72}$$

This leads to a spectrum $S_{\varepsilon\varepsilon}(\omega)$ having the same integrated value, and therefore corresponding to the same velocity variance,
as $\widehat{S}_{\nu\nu}(\omega)$, but that has no preference for positive or negative rotations. Thus rotationally anisotropic peaks, such as those
associated with coherent eddies or inertial oscillations, do not occur in $S_{\varepsilon\varepsilon}(\omega)$.

The basic idea is that since eddies and other oscillatory components will tend to raise the spectral levels above that due to
the background, taking the minimum tends to isolate velocities associated with the background displacement process $\mathbf{x}_\epsilon(t)$. At
the same time, the spectrum itself is a random quantity, and thus by always choosing the minimum we are guaranteeing that we
will underestimate the true level of an isotropic spectrum. Therefore it is necessary to use $c_\varepsilon$ to raise the overall level of $S_{\varepsilon\varepsilon}(\omega)$
somewhat, preserving its shape, to ensure that $S_{\varepsilon\varepsilon}(\omega)$ and $\widehat{S}_{\nu\nu}(\omega)$ integrate to the same value and therefore correspond to the
same velocity variance.

It is straightforward now to create a stochastic velocity timeseries $\varepsilon(t)$ having identically this spectrum. One simply takes
the square root of $S_{\varepsilon\varepsilon}(\omega)$, multiplies it by an array of unit-variance complex-valued Gaussian white noise, and inverse Fourier
transforms the result. Then to $\varepsilon(t)$ we add back the observed temporal mean velocity that was subtracted earlier from $\nu(t)$, and
finally we integrate the result on the sphere from the initial position of the original trajectory, leading to stochastic trajectories
with the specified velocities. One may show that these velocities are anisotropic in physical space as well as in terms of their
rotary components.

Proceeding in this way for each trajectory, we obtain a dataset that is the same size as the original dataset. Trajectory by
trajectory, the timeseries duration, initial location, temporal mean velocity,[7] velocity variance, and approximate spectral form

---

[7]Note that the match between the temporal mean velocities obtained by differencing the trajectories is approximate rather than exact, due to minor
differences between differentiating trajectories and integrating velocities on the sphere in the numerical implementation used here.

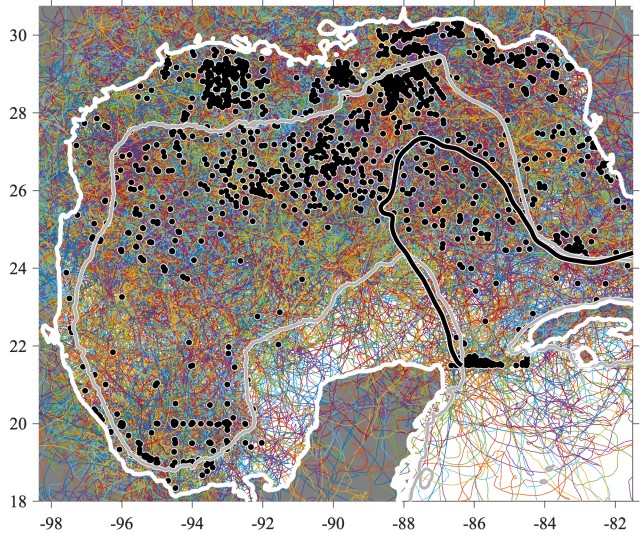

**Figure 11.** As in Fig. 1 but for trajectories of a noise dataset the same size as the original dataset. As described in the text, the noise is constructed on a trajectory-by-trajectory basis to have matching first- and second-order statistics as well as a similar velocity spectrum, with the exception than the velocity spectrum is constrained to be isotropic. The heavy white contour is the coastline.

all match by construction. One realization of this dataset is shown in Fig. 11. Comparison with Fig. 1 shows that the noise
730 dataset has a comparable visual degree of roughness to the original, a consequence of having matched the spectral shape. The gyre-scale circulation and tendency for systematic rotational motions are absent from the noise dataset, as intended. The fact that the noise trajectories traverse paths reaching outside the domain, although it may "look wrong", is irrelevant for our application; constraining particles to remain within the oceanic domain has no effect in improve the stochastic model for our purposes.

735 **4.4 Physical properties of ridges, and ridge averages**

To the various ellipse quantities introduced in Section 2, we will add several more that are particularly relevant for the study of coherent eddies. Firstly we note that just as there the displacement signal of a modulated oscillation describe an ellipse, so too does the associated velocity signal. Following Lilly and Gascard (2006), once the ellipse parameters are known for a displacement signal $\mathbf{x}_\star(t)$, the parameters of the ellipse on the velocity plane associated with $\mathbf{x}'_\star(t)$ can be found at once, see
740 Appendix E therein. The semi-axes of the velocity ellipse will be denoted $\tilde{a}(t)$ and $\tilde{b}(t)$. Then

$$R(t) \equiv \sqrt{a(t)b(t)} \tag{73}$$

$$V(t) \equiv \mathrm{sgn}\left(\tilde{b}(t)\right)\sqrt{\frac{\tilde{a}^2(t) + \tilde{b}^2(t)}{2}} \tag{74}$$





are respectively the instantaneous geometric mean radius and measure of the ellipse velocity. Here $V(t)$, which is a signed analogue of $\kappa$ for the velocity ellipse, is called the *kinetic energy velocity* since $V^2(t)$ gives the kinetic energy associated with the elliptical motion.

The quantities $V(t)$, $b(t)$, $\varsigma(t)$, and $\widetilde{\omega}(t)$ can all be of either sign. The former three are all positive for counterclockwise motion and negative for clockwise motion, whereas $\widetilde{\omega}(t)$ is positive for cyclonic motion and negative for anticyclonic motion. This distinction is not important for our northern hemisphere study, but is relevant for a global application.

To compactly summarize the results of the ridge analysis, we will look at properties averaged along the duration of a ridge. Denote the time average of some quantity $q(t)$ along a ridge by

$$\langle q \rangle = \frac{1}{t_f - t_i} \int\limits_{t_i}^{t_f} q(t) \mathrm{d}t \tag{75}$$

where times $t_i$ and $t_f$ mark the start and end of the ridge. Then a set of aggregate quantities representing average properties along a ridge is, recalling $\kappa \equiv \sqrt{[a^2(t) + b^2(t)]/2}$,

$$R_\star \equiv \sqrt{\langle R^2 \rangle}, \qquad \varsigma_\star \equiv \frac{\langle \kappa^2 \varsigma \rangle}{\langle \kappa^2 \rangle} \tag{76}$$

$$V_\star \equiv \mathrm{sgn}(\varsigma_\star) \sqrt{\langle V^2 \rangle}, \qquad \widetilde{\omega}_\star \equiv \frac{\langle \kappa^2 \widetilde{\omega} \rangle}{\langle \kappa^2 \rangle} \tag{77}$$

where care has been taken to form the average in an appropriate way for each quantity. For example, $V_\star^2$ is the time-averaged kinetic energy associated with the oscillation. The averages for $\varsigma_\star$ and $\widetilde{\omega}_\star$ are temporal averages weighted by the instantaneous signal power. The quantities $\varsigma_\star$, $V_\star$, and $\widetilde{\omega}_\star$, are all signed, reflecting the direction of orbit around the ellipse.

### 4.5 Ridge analysis initial results

The results of applying the ridge analysis using the settings described above to both the surface drifter dataset, and to the noise dataset, is summarized in Fig. 12 through scatter plots of ridge-averaged quantities. The real-world data is shown at the left and the noise dataset is shown at the right, with the $x$-axis being the ridge-averaged geometric mean radius $R_\star$ in all panels. In the data, 14469 ridges are found in the data with $L_\star \geq 2\sqrt{6}$. It turns out that the vast majority of these are due to inertial oscillations, with only 2511 ridges having $\widetilde{\omega}_\star > -1/2$. The noise dataset has 2025 ridges, comparable to the number of non-inertial ridges in the drifter dataset, and reflecting the problem of false positives.

A number of important features in the data are apparent in this figure. First, there is large concentration of long-duration ridges with nondimensional frequencies $\widetilde{\omega}_\star \approx -1$, due to inertial oscillations. Second, there is a major asymmetry between cyclonic and anticyclonic events, as is most apparent from Fig. 12c. From $R_\star = 0$ km all the way up until about $R_\star = 50$ km, large-amplitude, long-duration cyclonic events are observed that have no parallel on the anticyclonic side. At still larger radii, the asymmetry reverses, and there is a tendency for larger-amplitude, longer-duration anticyclonic events that do not occur on the cyclonic side, as is more clear from Fig. 12a. As discussed more later, these are the large eddies shed from the Loop Current (e.g Elliott, 1982; Lipphardt Jr. et al., 2008; Hall and Leben, 2016).





**Figure 12.** Scatter plots of ridge-averaged quantities from the surface drifter data, at left, together with the same plots for a noise dataset of the same size, at right. In (a) and (b), the location of each ridge in radius/velocity space is shown, by plotting the geometric mean radius $R_\star$ against the kinetic energy velocity $V_\star$. The color and size both represent the ridge length $L_\star$, defined to be the number of periods an oscillation executes along the ridge. In (c) and (d), the y-axis is now the nondimensional frequency $\widetilde{\omega}_\star$, while in (e) and (f) it is the circularity

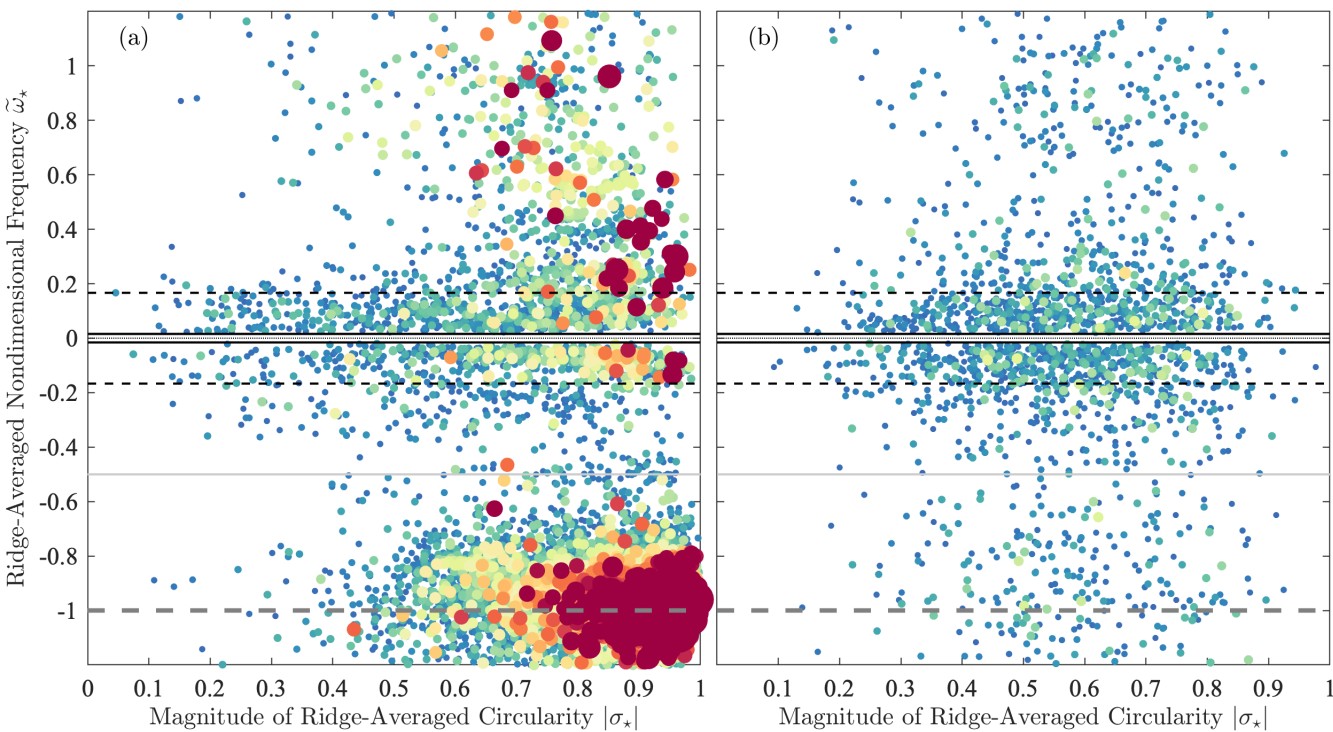

**Figure 13.** As in the lower row of Fig. 12, with the nondimensional frequency $\widetilde{\omega}_\star$ on the y-axis, but this time with the magnitude of the ridge-averaged circularity $|\varsigma_\star|$ on the x-axis. Again the real-world data is on left in (a) and the noise dataset is on the right in (b).

Another presentation of the ridge-averaged properties is that in Fig. 13, in which the $y$-axis is the nondimensional frequency $\widetilde{\omega}_\star$, while the $x$-axis is the magnitude of the circularity $|\varsigma_\star|$. These correspond, respectively, to the $y$-axes in the second and
third rows of Fig. 12. Here, the very high concentration of inertial oscillation ridges is apparent; these are too stacked on top of each other in Fig. 12 to stand out. A gap occurs, in the range of frequencies between about $\widetilde{\omega}_\star = -0.3$ and $\widetilde{\omega}_\star = -0.6$, where few events are seen. This gap, which does not occur on the cyclonic side, is most likely a manifestation of the stability boundary at $\widetilde{\omega}_\star = -1/2$, which as discussed previously affects only anticyclones. It is striking to see this dynamical feature emerge so readily from a data analysis methodology that makes no dynamical assumptions.

The inertial oscillations, as expected, are seen to be strongly circularly polarized in Fig. 13a. Outside of the inertial band, one observes a tendency for longer-duration events to be more strongly circularly polarized than shorter-duration events. On the cyclonic side a transition occurs around $\widetilde{\omega}_\star = 1/2$. The longest-duration cyclonic events at higher frequencies, while still highly circularly polarized, have somewhat lower degrees of circularly polarization than the longest-duration cyclonic events at smaller frequencies. This is curious because one might expect high-frequency, highly nonlinear events to also be circularly
polarized.

The right-hand columns of these two figures show the properties of the ridge analysis applied to the noise dataset seen in Fig. 11. In comparing the two columns, one sees that the real-world data contains ridges that have higher velocities, higher





frequency magnitudes and thus higher degrees of nonlinearity, and stronger degrees of circular polarization than those occurring in the noise. In addition, ridge lengths $L_\star$ in the data are observed to be frequently much larger than in the noise. While there

are 312 events in the data above the inertial band with lengths $L_\star > 4$, only sixteen such events occur in the noise. However, one also sees that low frequency, weakly circularly polarized events in the data align well with the noise distribution in both Fig. 12 and Fig. 13, suggesting that such events may not be statistically significant.

A final issue that should be mentioned is the possibility of contamination by the tides. The semidiurnal tide is excluded by our chosen frequency range, $\widetilde{\omega} = 1/64$–$2$, except in the very northernmost part of the domain; the $M2$ frequency exceeds twice

the local Coriolis frequency above about $28.7°$ N. Aliasing of semi-diurnal tidal motions into eddy-like signals is therefore not expected to be a significant problem. The diurnal tide, meanwhile, is close to the inertial frequency over the range of latitudes spanned by the Gulf, and ranges from about $1.6f$ at $18°$ N to $1.0f$ at $30°$ N. Thus, the band where we might see diurnal tidal motions is $\widetilde{\omega} = 1$–$1.6$. A conspicuous tendency of oscillations at the diurnal frequency is not observed (not shown) and therefore does not appear to a major source of eddy-like variability. This topic could, however, be given more attention by

examining the spectral signals of detected oscillations, which are far more narrowband for tidal signals than for eddies.

### 4.6   Assessing statistical significance

The plots from the previous section emphasize the importance of excluding false positives through an assessment of statistical significance. In doing so, it is desirable to avoid *ad hoc* cutoffs that involve the very properties we are most interested in studying, such as the radius, velocity, or Rossby number. If we consider what is the essence of coherent vortices, we can say

that they are (i) long-lived features by definition and (ii) roughly circular, with the possibility of small degrees of eccentricity arising due to various dynamical processes such as ambient strain. Thus suggests that $L_\star$ and $|\varsigma_\star|$ would be natural measures of significance that are agnostic to the primary physical properties of the vortex. The last two figures show that indeed, the data and noise do have different distributions with respect to these two quantities.

Importantly, both $L_\star$ and $|\varsigma_\star|$ have distributions in the noise that appear to be invariant to the ridge mean frequency $\widetilde{\omega}_\star$. In 40

bins of width 0.1 between $\widetilde{\omega}_\star = -2$ and $\widetilde{\omega}_\star = 2$—the minimum and maxima frequency employed in the wavelet transform— and neglecting the outermost bins where only a single event is observed in the noise, we find the mean value of $L_\star$ in the noise is $2.02 \pm 0.10$ cycles, while the mean value of $|\varsigma_\star|$ is $0.59 \pm 0.02$, where the quantity after the $\pm$ is the standard deviation across bins. Thus, the mean values do not exhibit a frequency dependence. Similarly, the standard deviation of $L_\star$ within a bin is $0.48 \pm 0.14$ cycles while that of $|\varsigma_\star|$ and $0.16 \pm 0.01$, so these also appear invariant to frequency. The fact that the distributions

of $L_\star$ and $|\varsigma_\star|$ are invariant to frequency in the noise indicates that it is not necessary to separately test for significance in different frequency bins, a considerable simplification.

A novel significance criterion is created as follows. Assume that the probability distribution of the observed ridges on the $\varsigma_\star / L_\star$ plane is proportional to the function

$$g_\star\left(\varsigma_\star, L_\star\right) \tag{78}$$





using "$g$" rather than the standard notation "$f$" because that symbol is already being used for the Coriolis frequency. Whereas probability distributions are defined to integrate to one, here we will make a different choice. The double integral of $g_\star(\varsigma_\star, L_\star)$ is normalized such that

$$\int\limits_{-\infty}^{\infty} \int\limits_{\infty}^{\infty} g_\star(x,y)\,\mathrm{d}x\mathrm{d}y = \frac{\text{ridge points}}{\text{data points}} \tag{79}$$

is the ratio of the total number of ridge points to the total number of data points in an entire dataset, accounting for the possibility of multiplicities greater than one. In other words, this integral gives average number of ridge points per data point, or *ridge density*. Note, this is not quite the same as the probability of data point being a ridge point, because a data point could be a member of more than one ridge.

The familiar cumulative distribution function corresponding to $g_\star(x,y)$ is obtained by a double integration on the $\varsigma_\star/L_\star$ plane up to a particular $(\varsigma_\star, L_\star)$ value

$$G_\star(\varsigma_\star, L_\star) \equiv \int\limits_{-\infty}^{\varsigma_\star} \int\limits_{-\infty}^{L_\star} g_\star(x,y)\,\mathrm{d}x\mathrm{d}y. \tag{80}$$

Due to our choice of normalization, $G_\star(\varsigma_\star, L_\star)$ at any $(\varsigma_\star, L_\star)$ point is the value of the ridge density considering all ridge points having $\varsigma_\star$ and $L_\star$ both simultaneously less than the given values. Doubly integrating the density function in the opposite directions leads to a less familiar function

$$\mathcal{S}_\star(\varsigma_\star, L_\star) \equiv \int\limits_{\varsigma_\star}^{\infty} \int\limits_{L_\star}^{\infty} g_\star(x,y)\,\mathrm{d}x\mathrm{d}y \tag{81}$$

which is known as the *complementary cumulatively distribution function* or, more colorfully, the *survival function*. $\mathcal{S}_\star(\varsigma_\star, L_\star)$ gives the ridge density for all ridge points having $\varsigma_\star$ and $L_\star$ both simultaneously *greater* than the given values.

Comparing the estimated survival function of the data to that of the noise will allow us to assess whether an event's properties are extreme enough to warrant classifying it as statistically significant. The construction of our measure of statistical significance is described with reference to Fig. 14. Here, we exclude ridges with $\widetilde{\omega}_\star < -1/2$ from both the data and the noise, which as discussed above will remove all inertial oscillations. In panel (a), a black dot is placed marking the properties of each ridge in the data, and in (b) the same is done for the noise dataset. These dots are the discrete, sampled expressions of the $\varsigma_\star/L_\star$ density function of the data $g_\star(\varsigma_\star, L_\star)$ and also that of the noise, denoted $g_\varepsilon(\varsigma_\star, L_\star)$.

The density functions $g_\star(\varsigma_\star, L_\star)$ and $g_\varepsilon(\varsigma_\star, L_\star)$ themselves are then estimated by forming two-dimensional histograms on the $\varsigma_\star/L_\star$ plane. For better resolution of the noise histogram, it is computed using ten different realizations, then divided by ten to ensure that it reflects the average properties of a dataset of the same size as the original. Cumulatively summing these two histograms along both axes, but from right-to-left and top-to-bottom instead of the more usual reverse directions, leads to estimates of the survival functions $\widehat{\mathcal{S}}_\star(\varsigma_\star, L_\star)$ and $\widehat{\mathcal{S}}_\varepsilon(\varsigma_\star, L_\star)$ for the data and the noise. These are shown as the colored shading in Fig. 14a and b.

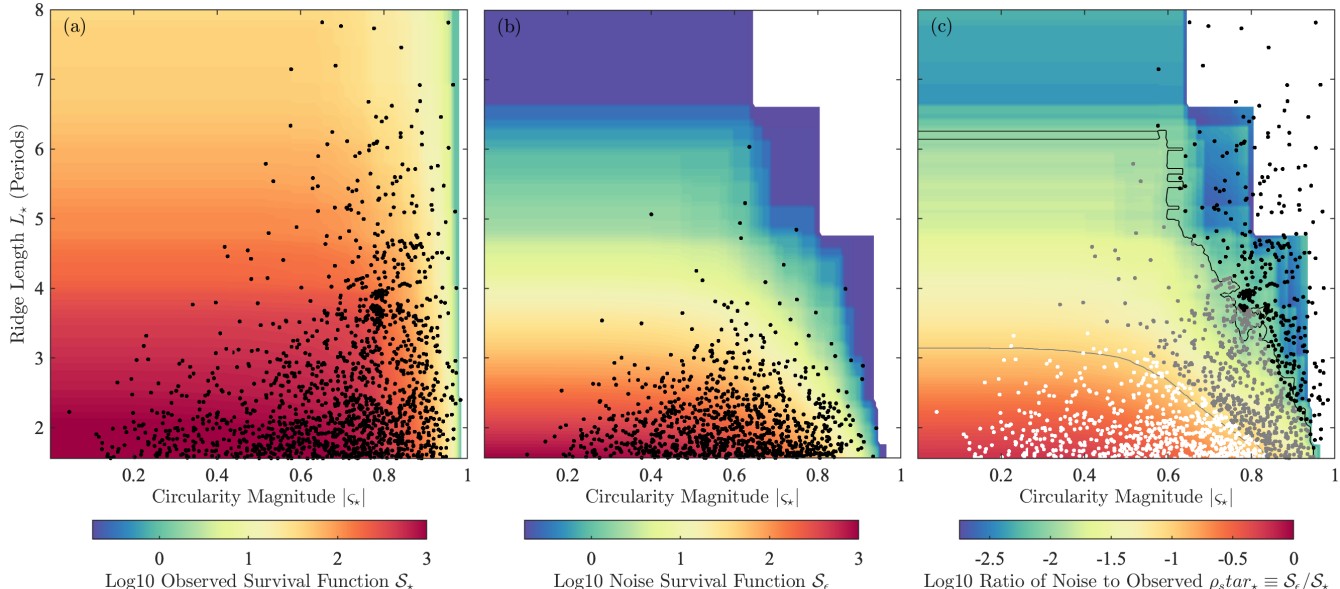

**Figure 14.** A significance criterion for wavelet ridges in the Gulf of Mexico surface drifter dataset. The black dots in panel (a) show the locations of all ridges in the ridge-averaged circularity magnitude $|\varsigma_\star|$ versus ridge duration $L_\star$ space. Bin-averaging to form the two-dimensional histogram, and then cumulatively summing this in both dimensions, but from high values to low values rather than the more usual low to high values, leads to an object called the complementary cumulative distribution function or survival function $\mathcal{S}_\star(\varsigma_\star, L_\star)$, shown as the colored shading in (a). Panel (b) is constructed in the same way but for the noise dataset. Since this dataset is ten times the size of the observations, for display purposes only 10% of the dots are shown, and the histogram is divided by a factor of 10, thus making these panels (a) and (b) directly comparable. Finally the logarithm of the ratio $\rho_\star(\varsigma_\star, L_\star) \equiv \mathcal{S}_\varepsilon/\mathcal{S}_\star$ is shown in (c). The gray line in this panel marks the $\rho(\varsigma_\star, L_\star) = 0.1$ contour such that only one out of ten events above or to the right of this curve is expected to be a false positive. Similarly, the black line is the $\rho(\varsigma_\star, L_\star) = 0.01$ contour, above which only one out of 100 events is expected to be a false positive. Dots in this panel are those of the data, the same as in (a). The gray and black dots mark observed ridges falling above the 0.1 and 0.01 contours, respectively. Those falling below the $\rho(\varsigma_\star, L_\star) = 0.1$ contour, shown as white dots, are judged to be statistically insignificant and are rejected.

The ratio of the estimated $\varsigma_\star/L_\star$ survival function of the noise to that of the data

$$\rho_\star(\varsigma_\star, L_\star) \equiv \frac{\widehat{\mathcal{S}}_\varepsilon(\varsigma_\star, L_\star)}{\widehat{\mathcal{S}}_\star(\varsigma_\star, L_\star)} \tag{82}$$

is a measure of statistical significance that will be called the *density ratio*, see Fig. 14c. Consider the contour $\rho_\star(\varsigma_\star, L_\star) = 0.1$, the thin gray line in this panel. Ridges with properties above or to the right of this line occur only ten percent as often in the noise dataset compared with the original data. Therefore, only one in ten events in the data with properties above this line is expected to be a false positive under the null hypothesis of the isotropic velocity spectrum. Similarly, events above the black contour at $\rho_\star(\varsigma_\star, L_\star) = 0.01$ occur only 1% as often in the noise as in the data, thus only one in one hundred is attributable to the noise spectrum.





In establishing a measure of statistical significance, we have refrained from using the language of statistical hypothesis testing, e.g. significance levels, $p$-values, etc. In statistical terminology, the probability under the null hypothesis of observing a value in a test statistic that is more extreme than a given observation is known as the $p$-value of that observation. The $\rho_\star$-value by contrast, gives the estimated probability that a detected event is actually a false positive has been incorrectly accepted; these are related but different. The $\rho_\star$-value could also be seen as being similar to the "false discovery rate" in multiple hypothesis testing introduced by Benjamini and Hochberg (1995). Because it seems natural to approach this problem in terms of the *densities* of detected events, our proposed measure of statistical significance differs from those commonly used in the hypothesis testing literature. Formally connecting these ideas would be a useful exercise, but is outside the scope of the present paper.

### 4.7 Results

Excluding now those ridges having a density ratio exceeding $\rho_\star(\varsigma_\star, L_\star) = 0.1$, the gray line in Fig. 14c, as well as those with nondimensional frequencies below $\widetilde{\omega}_\star < -1/2$ in order to exclude inertial oscillations, we obtain only 1390 statistically signficant ridges remaining, or about 69% of the 2029 non-inertial ridges. In the noise, by construction, there are only 139 ridges meeting these criteria in a typical dataset of the same size as the original dataset. The distribution of events from this statistically significant ridge set, termed the *eddy ridges*, will be briefly described in this section.

The ridge-averaged properties of the eddy ridges are shown in Fig. 15. The major feature is the asymmetry between cyclones and anticyclones noted in the Introduction. Three regimes can be noted: $R_\star < 10$ km, dominated by long-lived, very highly nonlinear cyclones, with only relatively weak and short-duration anticyclones; $10$ km $< R_\star < 50$ km, dominated by nonlinear and very long-lived cyclones with nondimensional frequencies $\widetilde{\omega}_\star$ often greater than $1/6$, again with only relatively brief anticyclones, and with a striking lack of anticyclones below $\widetilde{\omega}_\star = -1/6$; and $R_\star > 50$, which is increasingly dominated by long-lived anticyclones as one moves to larger radii.

The ellipses corresponding to the eddy ridges are those shown in the earlier Fig. 2. The ellipse inversion equations, Eqs. (29)–(32), are applied to the extracted signals, and the resulting ellipses are plotted with a temporal spacing equal to the estimated period $2\pi/|\hat{\omega}(t)|$. The center of each ellipse, with latitude $\widehat{\Phi}_\diamond^m(t)$ and longitude $\widehat{\Theta}_\diamond^m(t)$, is the estimated as the residual after subtracting the ridge from the full trajectory, see Eqs. (58) and (59). Ridges are colored by their nondimensional instantaneous frequency magnitude $|\widetilde{\omega}(t)|$, which when multiplied by two gives an estimate of the magnitude of the Rossby number of the oscillatory flow.

Overall there is a striking difference in geographic distribution between cyclones and anticyclones. Cyclones are concentrated in eastern, western, and southern Gulf, generally excluding the shallow shelf regions. The anticyclones are concentrated primarily over the Loop Current and due west of it, corresponding to the deepest part of the Gulf (see Fig. 1 of Lilly and Pérez-Brunius, 2020a), with a smaller number of events seen north or northwest of the Loop Current. The two patterns are observed to be generally complementary, with cyclonic events and anticyclonic events occurring in largely disjoint locations.

The distribution of anticyclonic events exhibits many large features with $R_\star > 50$. Although not apparent in the map, one may observe trains of large ellipses tending to form near the Loop Current, then drifting westward, filling the central latitudes of the Gulf. Such events are capturing the well-known Loop Current eddies (e.g Elliott, 1982; Lipphardt Jr. et al., 2008; Hall





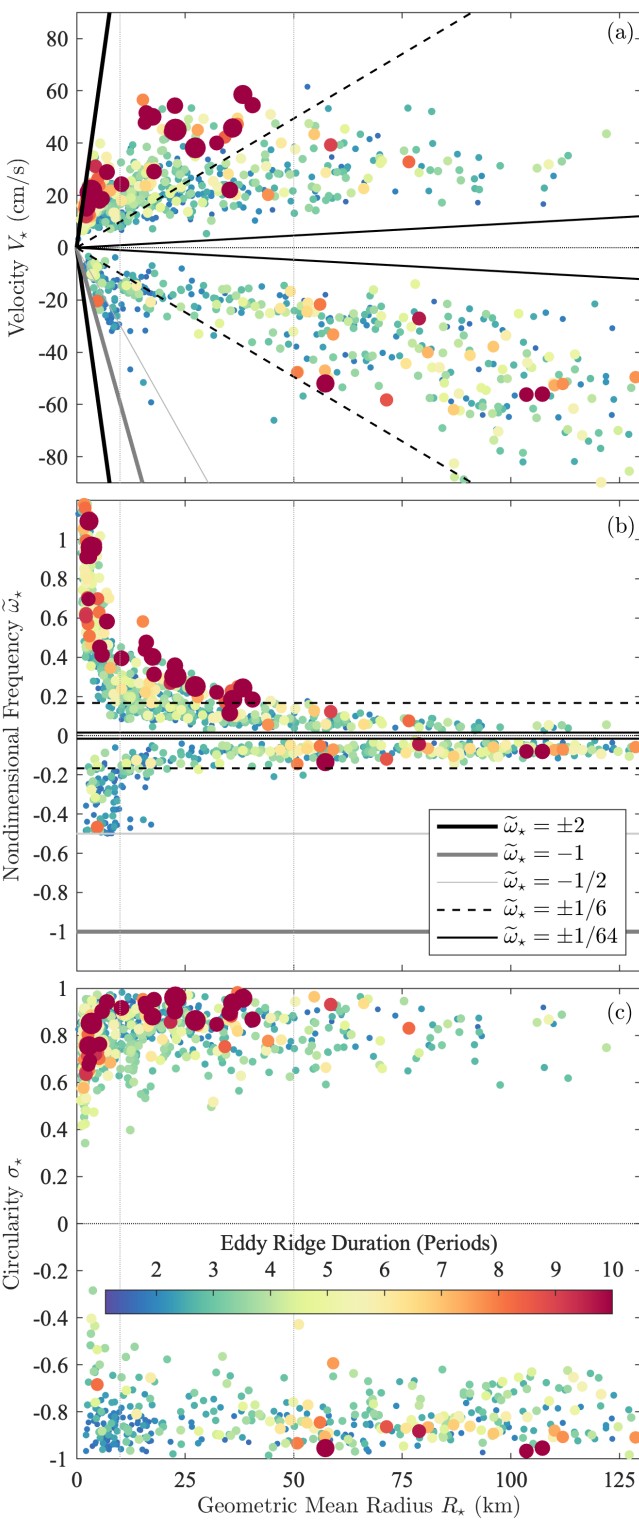

**Figure 15.** As in the left column of Fig. 12, but now showing the ridge-averaged properties of only the non-inertial ridges having a density ratio of $\rho_\star (\varsigma_\star, L_\star) \leq 0.1$, in other words, with at most one in ten events attributable to the null hypothesis of an isotropic background spectrum. Vertical lines at $R_\star = 10$ km and $R_\star = 50$ km denote apparent transitions between different regimes.





and Leben, 2016) that form periodically as pinch-off events from the Loop Current. From Fig. 15, especially in panel (a), one sees that there are many more anticyclones than cyclones for $R_\star > 50$ km, and especially for $R_\star > 100$ km. In this size class one also observes a number of long-lived, highly circular features on the anticyclonic side—the orange and red circles in Figs. 15a,c—but only a handful of similar features on the cyclonic side.

895    A cluster of cyclonic activity is seen in Fig. 2 to occur in the southwestern Gulf, a region known as the Bay of Campeche. The observed ellipses there span a range of sizes, including a concentration in the large size class with $R_\star > 50$ km. This is consistent with the known presence of a quasi-permanent vortex, the Campeche Gyre, in that region (Padilla-Pilotze, 1990; Vázquez De La Cerda et al., 2005; Pérez-Brunius et al., 2013), and suggests the eddy ellipses could be employed to usefully study its variability.

900    Two areas of energetic cyclonic eddy activity with roughly 10 to 50 km radius ellipses—the intermediate size class in Fig. 15—are the vicinity of the Loop Current and also the western Gulf. In both areas, one sees cyclonic ellipses with large nondimensional frequency magnitudes $|\widetilde{\omega}_\star|$, hence large Rossby numbers, as well as long durations $L_\star$. Note that such events are almost entirely absent from the anticyclonic distribution, as seen in Fig. 15b. These events correspond in the Fig. 2 to the yellow to orange colors, where we can see many ellipses of 50 km radius or less that often occur in long trains. In the east, these align with the periphery of the Loop Current, and may be identified with the so-called Loop Current Frontal Eddies or LCFEs that form in the strong cyclonic shear zone on the current's flank (e.g Le Hénaff et al., 2014). Cyclones with similar sizes and degrees of nonlinearity are observed in the western Gulf, with a notable absence in the central Gulf. Whether this is a real feature or the result of a sampling issue is not immediately clear. Cyclones are known to occur in the western Gulf, although most commonly in its northwest corner, see e.g. Merrell Jr. and Morrison (1981) and Hamilton et al. (2002). The similarity with LCFE's suggests investigating the possibility of a similar origin for the western eddies, either in the shear zone of the Loop Current or in those of the Loop Current eddies.

Finally, the small-radius ($R_\star < 10$ km), highly nonlinear cyclones seen in the scatter plot of Fig. 15 are visible as the small red circles broadly distributed throughout the Gulf. Far fewer events in this size class occur on the anticyclonic side. In particular, events with $|\widetilde{\omega}_\star| > 1/2$ are populous on the cyclonic side, but dynamically forbidden (and thus excluded from the census) on the anticyclonic side. While it is well known through remote sensing and sun-glint photographs that small cyclonic eddies are ubiquitous in the upper ocean (Munk et al., 2000; Eldevik and Dysthe, 2002), *in situ* observations of these so-called 'spiral eddies' have proved elusive. See Lumpkin and Elipot (2010) for a rare example in the literature of a submesoscale vortex observed through surface drifters, their Fig. 6, due to "an exceptional and extremely anomalous case" of two drifters trapped in the same 10 km eddy. The results here, in which we can detect submesoscale features in individual trajectories, suggests that the fact that they are not more commonly seen in drifter trajectories is the result of a technical limitation in existing analysis methods, and not because they are not present or not resolved.





## 5 Conclusions

This paper has presented a method for extracting the displacement signals associated with coherent eddies, and for estimating the properties of the features that generated those signals, from large Lagrangian datasets. The method is rooted in ideas
taken from the signal analysis literature, such as the analytic signal, modulated oscillations, and wavelet analysis. As these ideas are not yet all widely known in oceanography, they were introduced in some detail and discussed in the context of the eddy-detection problem.

A modification of an existing analysis method, multivariate wavelet ridge analysis, to prohibit sign transitions between cyclonic and anticyclonic polarizations renders it more suitable for the eddy-detection problem. The main innovation, however,
is a means of determining statistical significance, which is done through the creation of a null hypothesis in the form of a 'noise' dataset constructed to match the basic statistical properties of the real-world data, yet lacking organized oscillatory features arising from spectral peaks. A two-parameter criterion is proposed in which the joint distribution of (i) event duration, measured by the number of oscillations executed, and (ii) the degree of circular polarization is constructed for the data, and compared with that of the noise in order to identify parameter regimes in which the noise is unlikely to have generated the observed
features.

The statistically significant features emerging from an application to a large surface drifter dataset from the Gulf of Mexico were briefly discussed. It is clear that there is much to be learned about the Gulf of Mexico eddy field through studying these eddy ellipses. Here, in order to maintain a focus on the analysis method, attention paid to physical results is intentionally kept to a minimum. These will be thoroughly investigated in a follow-on paper.

The incorporation of a criterion for determining statistical significance makes possible the application of the eddy-extraction method to very large datasets. This has great potential not only for studying eddies in real-world data with a new level of detail, but also for providing a novel means of comparing eddy characteristics from numerical models with those from observations. In particular, the results from the Gulf of Mexico indicated that large-scale studies of submesoscale cyclonic vortices from *in situ* data are now possible for the first time. The distribution of a self-contained analysis routine, discussed below, is aimed to
help facilitate the usage of this method by other investigators.

It is intended that this paper provide the groundwork for a multi-paper effort to solve the eddy extraction and property estimation problem as completely and rigorously as possible. Future efforts will include (i) examining the behavior of the method with regard to exact analytic vortex solutions such as those reviewed in Lilly (2018); (ii) exploring the potential of ridge analysis in the light of the deep connection established by Lilly (2018) between Lagrangian ellipse properties and the
kinematic flow properties of vorticity, strain, and divergence; (iii) understanding the bias estimate of Lilly and Olhede (2012a) in terms of eddy dynamics through applications to idealized numerical models as well as to analytic solutions; and finally (iv) connecting this Lagrangian *trajectory* perspective to the rigorous Lagrangian *field* theory of Haller and Beron-Vera (2012) and related works.





*Code availability.*  A freely available software package

All numerical code required for the analysis carried out in this paper is distributed to the community as a part of jLab, the lead author's open source data analysis toolbox for Matlab, available at https://github.com/jonathanlilly/jLab and with extensive documentation found at http://www.jmlilly.net/doc/jLab.html. The centerpiece needed for this work is a new function, `eddyridges`, that implements the one-sided multivariate wavelet ridge analysis on an entire multi-instrument Lagrangian dataset. This may be applied either to latitude/longitude trajectories, as we have here, or to trajectories on the $x/y$ plane

such as may be output from a numerical model. This function extends and greatly simplifies the approach used in Lilly and Gascard (2006) and Lilly et al. (2011), which involved directly calling the lower-level `ridgewalk` function implementing the ridge analysis. Construction of the noise dataset is done through the use of the function `noisedrifters`. Assorted other functions for computing ellipse properties and so forth are also included; see the `jEllipses` and `jOceans` modules. Finally, the toolbox contains scripts for creating the `gomed.nc` dataset, `make_gomed`, and for generating all figures in this paper,

`makefigs_gulfcensus`.

    Whereas for simplicity the handling of spherical geometry has been discussed herein using a small angle expansion about a fixed point, the numerical implementation in `eddyridges` takes a different approach that offers better performance for trajectories covering large distances on the surface of the earth. This algorithm, using a routine called `spheretrans`, will be briefly discussed here. First, latitudes and longitudes are converted to a position in three-dimensional space, and the wavelet

transform of that displacement signal in three dimensions is taken. The wavelet transform is then projected back onto a plane tangent to the earth, centered on the time-varying center of the oscillation in each band of the transform considered separately. The multivariate ridge analysis is then applied to the resulting bivariate wavelet transform vector. In this way, the method uses a projection about a moving point that is suitable for each oscillation, rather than a single fixed point for an entire dataset.

*Data availability.*  The Gulf of Mexico Eddy Dataset (GOMED)

The dataset utilized in this paper is the consolidated surface drifter dataset created by Lilly and Pérez-Brunius (2020a) and referred to as GulfDriftersAll therein. While a subset of this data is proprietary and unfortunately cannot be redistributed, the bulk of that dataset is publicly-available and is released to the community as the NetCDF file GulfDriftersFree, available at https://doi.org/10.5281/zenodo.3985916 (Lilly and Pérez-Brunius, 2020b).

    The results of applying the multivariate wavelet ridge analysis to GulfDriftersAll is distributed to the community as the Gulf

of Mexico Eddy Dataset (GOMED) at https://10.5281/zenodo.3978803 (Lilly and Pérez-Brunius, 2020c). Table A1 provides an overview of all variables contained in GOMED, which is distributed as the NetCDF file `gomed.nc`. Variables include extracted eddy displacement signals for all ridges, significant or not, detected using the settings described herein, as well as the time-varying ellipse parameters and estimated ellipse center location. The data includes eddy displacement signals for all ridges, as well as the time-varying ellipse parameters and estimated ellipse center location. The instantaneous frequency is

also included, as is the instantaneous bias estimate derived by Lilly and Olhede (2012). The data are organized as appended





trajectory data that can be readily separated through the use of the `ids` field. The ridge length $L_\star$ and ridge-averaged circularity $\varsigma_\star$ are also included, as is the measure of statistical significance $\rho_\star$.

**Appendix A: Construction of the synthetic signal**

The synthetic signal shown in Fig. 4a is formed using the ellipse generation equation, Eq. (1), with parameters

$$\phi(t) = \omega_o t \left(1 + t/T\right) \tag{A1}$$

$$\theta(t) = \frac{\pi}{2} - \omega_o t/10 \tag{A2}$$

$$\kappa(t) = \kappa_o \left(1 + 5t/T\right) \tag{A3}$$

$$\varsigma(t) = - \begin{cases} 1 & t < T/4 \\ \sqrt{1 - \left[\frac{1}{3}\left(4t/T - 1\right)\right]^2} & t \geq T/4 \end{cases} \tag{A4}$$

where $\omega_o = 0.05$ rad day$^{-1}$, $\kappa_o = 10$ km, and $T = 1000$ days, the latter being the signal duration. The semi-axes lengths are

recovered from $\kappa(t)$ and $\varsigma(t)$ via Eq. (33) and (34). These choices lead to an clockwise circulating ellipse with a linearly growing amplitude, $\kappa(t)$, that increases sixfold over the signal duration $T$, and also a linearly increasing orbital frequency, $\omega_\phi(t) = \phi'(t) = (1 + 2t/T)\omega_o$, that doubles. The ellipse is purely circular for the first quarter of its duration, $|\varsigma(t)| = 1$. After that, it begins to become increasingly elliptical, reaching a vanishing circularity $\varsigma(t) = 0$ at $t = T$, while steadily precessing in the clockwise direction at a rate that is 10% of its initial orbital frequency, $\omega_\theta(t) = \theta'(t) = -\omega_o/10$.

The composite signal shown in Fig. 5 is constructed by adding the elliptical signal generated above to (i) a uniform westward drift at $0.5$ cm s$^{-1}$ plus (ii) a stochastic component. For the latter, we use a Matérn velocity process, shown by Lilly et al. (2017) to be equivalent to damped fractional Brownian motion. Referring to Eq. (47) therein, the velocity standard deviation $\sigma$ is set to $\sigma = 0.25$ cm s$^{-1}$, the spectral slope parameter $\alpha$ is chosen as $\alpha = 1$ giving a high-frequency decay of $\omega^{-2}$, and the damping parameter $\lambda$ is set to $\lambda = 0.1$ days$^{-1}$. This velocity process is then cumulatively summed to generate a displacement signal.

The time-varying period corresponding to this signal, shown in Fig. 8b as the thin gray line, is computed by differentiating the generating ellipse parameters as $2\pi / \left[\phi'(t) + \varsigma(t)\,\theta'(t)\right]$. This is based on the form of Eq. (37) for the instantaneous frequency in terms of the canonical ellipse parameters.

*Author contributions.* JML was responsible for the theory, coding, analysis, and writing. PPB was responsible for obtaining funding, for the planning, deployment, and upstream processing of several of the datasets utilized herein, for managing the sharing of the GOMED dataset,
and for finding the legal pathway to make this dataset publicly available. She also provided regional expertise and guidance throughout this project.

*Competing interests.* The authors declare no competing interests.



Symbols used in this paper and in GOMED dataset

| Symbol | Variable | Meaning | |
|---|---|---|---|
| *Wavelet-related symbols* | | | |
| $\beta,\gamma$ | — | Wavelet order parameter $\beta$ and shape parameter $\gamma$, set here to $\gamma = 3$ | Eq. (42) |
| $\psi_{\beta,\gamma}(t)$ | — | Time-domain generalized Morse wavelet | Eq. (44) |
| $\Psi_{\beta,\gamma}(\omega)$ | — | Frequency-domain generalized Morse wavelet | Eq. (42) |
| $\omega_{\beta,\gamma}$ | — | 'Peak frequency' $\omega_{\beta,\gamma} \equiv (\beta/\gamma)^{1/\gamma}$ at which $\left|\Psi_{\beta,\gamma}(\omega)\right|$ is maximized | Eq. (44) |
| $P_{\beta,\gamma}$ | — | Nondimensional wavelet duration $P_{\beta,\gamma} = \sqrt{\beta\gamma}$ | Eq. (45) |
| $a_{\beta,\gamma}$ | — | Normalizing constant $a_{\beta,\gamma} \equiv 2(\mathrm{e}\gamma/\beta)^{\beta/\gamma}$ giving $\Psi_{\beta,\gamma}(\omega_{\beta,\gamma}) = 2$ | Eq. (43) |
| $w(t,s)$ | — | Wavelet transform of a univariate signal | Eqs. (47)–(48) |
| $\mathbf{w}(t,s)$ | — | Wavelet transform of a vector-valued signal | Eq. (52) |
| *Per-ridge quantities* | | | |
| — | `ridge_id` | Ridge ID number | — |
| — | `segment_id` | ID of segment number from within GulfDriftersAll | — |
| — | `drifter_id` | ID of originating drifter from within GulfDriftersAll | — |
| — | `row_size` | Number of observation points within each ridge | — |
| $\rho_\star$ | `rho_star` | Ridge significance level | Eq. (82) |
| $\varsigma_\star$ | `sigma_star` | Power-weighted ridge-averaged circularity $\varsigma(t)$ | Eq. (76) |
| $L_\star$ | `L_star` | Ridge length in number of oscillations | Eq. (62) |
| *Along-ridge timeseries* | | | |
| — | `ids` | Ridge ID number, same as `ridge_id`, repeated for all data points | — |
| $t$ | `time` | Time in days since 00/00/0000 | — |
| $\Phi(t)$ | `lat` | Latitude of trajectory segment during ridge | Eq. (38)&(40) |
| $\Theta(t)$ | `lon` | Longitude of trajectory segment during ridge | Eq. (38)&(41) |
| $\widehat{\Phi}_\diamond(t)$ | `latres` | Estimated time-varying central latitude of ridge | Eq. (58) |
| $\widehat{\Theta}_\diamond(t)$ | `lonres` | Estimated time-varying central longitude of ridge | Eq. (59) |
| $x_\star(t)$ | `x_star` | Eastward displacement associated with eddy ellipse | Eqs. (39) &(55) |
| $y_\star(t)$ | `y_star` | Northward displacement associated with eddy ellipse | Eqs. (39) &(55) |
| $\kappa(t)$ | `kappa` | Root-mean-square ellipse radius $\kappa \equiv \sqrt{[a^2(t)+b^2(t)]/2}$ | Eq. (5) |
| $\varsigma(t)$ | `sigma` | Ellipse circularity $\varsigma(t) \equiv 2a(t)b(t)/\left[a^2(t)+b^2(t)\right]$, $|\varsigma(t)| \leq 1$ | Eq. (5) |
| $\theta(t)$ | `theta` | Orientation angle of ellipse major axis counterclockwise from $x$-axis | Eq. (1) |
| $\phi(t)$ | `phi` | Phase angle indicating particle location with respect to major axis | Eq. (1) |
| $\omega(t)$ | `omega` | Instantaneous frequency, given by $\omega(t) = \phi'(t) + \varsigma(t)\theta'(t)$ | Eqs. (35)–(37) |
| $\widetilde{\omega}(t)$ | `omega_tilde` | Nondimensional instantaneous frequency, $\widetilde{\omega}(t) \equiv \mathrm{sgn}\left(\varsigma(t)\right)\omega(t)/f_\diamond(t)$ | Eq. (65) |
| $R(t)$ | `R` | Geometric mean radius $R(t) \equiv \sqrt{a(t)b(t)}$ | Eq. (73) |
| $V(t)$ | `V` | Ellipse kinetic energy velocity, $\mathrm{sgn}\left(V(t)\right) = \mathrm{sgn}\left(b(t)\right)$ | Eq. (74) |
| $\chi(t)$ | `chi` | Estimated nondimensional bias error, Eq. (62) of Lilly and Olhede (2012a) | — |

**Table A1.** Some important symbols used in this paper, together with variables from the GOMED dataset. The quantities in the second and third sections of the table are all the variables appearing in the GOMED dataset.





*Acknowledgements.* This research has been partially funded by the Mexican National Council for Science and Technology - Mexican Ministry of Energy - Hydrocarbon Fund, project 201441. This is a contribution of the Gulf of Mexico Research Consortium (CIGoM). We acknowledge the specific request of Petróleos Mexicano (Pemex) to the Hydrocarbon Fund to address the environmental effects of oil spills in the Gulf of Mexico that made this project possible. The work of JML was supported in part by award #1658564 from the Physical Oceanography program of the United States National Science Foundation.





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
