# Peer review of "Extracting statistically significant eddy signals from large Lagrangian datasets using wavelet ridge analysis, with application to the Gulf of Mexico"

_Nonlinear Processes in Geophysics, 2020_

## Referee Comment (RC1) · Anonymous Referee #1 · 20 Oct 2020

October 2020 review of:

"Extracting statistically significant eddy signals from large Lagrangian datasets using wavelet ridge analysis, with application to the Gulf of Mexico" by Jonathan M. Lilly and Paula Pérez-Brunius

General comments:

This is a methodological paper about eddy detection in oceanic surface drifter data from the Gulf of Mexico. It builds on previous methodology, esp. by the first author.

[Figure]

The focus is on refinement of the methodology, as well as documenting the underlying theoretical principles. Only a small portion of the paper addresses the oceanographic findings about the Gulf of Mexico circulation.

The strengths of this paper include its thorough presentation of the mathematical theory behind the method. It is also very well-written and presented in a clear, logical order. I have no doubt that the method is sound, well-understood, and well documented. The application to the observational data from the Gulf of Mexico shows that it works and delivers results.

However, the paper does have weaknesses:

Given that a number of methods with similar objectives already exist, it is not really clear why another one is needed. Other methods might not be as elegant or well-documented, but does this alone really justify an entire paper? The case would be clear if it were shown that existing methods deliver wrong or ambiguous results, or as a minimum, if there were a table that showed each method's capabilities and short-comings. The authors mention the existence of earlier methods (e.g. references from Dong et al., 2011), and claim that these methods are problematic to apply to the given dataset. I find this claim unsubstantiated, and possibly incorrect, but have to admit that I have not tried any of the methods myself here. Therefore, this paper is an incremental improvement from the previously existing method by the first author - no more, no less. To my knowledge, the attempt to quantify the background flow and thereby the statistical significance of the detected eddy numbers is novel. This is a valuable addition to the method.

At 51 pages, the paper is very long. I was going to recommend significant shortening, but have since realized that the text reads very fluently. It would not help to have the same content at half the length, but having to read every paragraph twice to understand it. Therefore, I will not recommend drastic shortening here. If some minor shortening is desired, e.g. to make room to discuss the other eddy methods, figures 11 and 13 and

their discussions could be descoped or downgraded into an appendix.

The paper purposefully keeps the oceanographic results to a minimum, and announces a future manuscript to follow up. This is appropriate for a methodology paper, but a bit more context should be provided: Have other Lagrangian eddy detection methods reported the same asymmetry between cyclones and anticyclones, or is this asymmetry a new finding or specific to the Gulf of Mexico? Are similar eddy properties, incl. the asymmetries between cyclones and anticyclones, generally present in eddy-resolving numerical circulation models?

There is a large number of self-citations in the references. While it is a good thing to provide background information, a malevolent interpretation could be that not many people other than the authors themselves care about this topic. This makes the paper appear less significant. As of Oct. 19, 2020, there were no public discussion comments about the online version of the paper. This, too, reflects poorly on the significance of the paper. Perhaps the authors can still ask a handfull of peers to submit comments before the deadline? This could at least show that interested readers exist.

In the introduction section, the word "dynamic" is used in a way that does not match my expectations. I was expecting "dynamics" to refer to explanations (forces, energy budgets, Navier-Stokes equations) behind the observed motions, and would distinguish this from "kinematics", the mere description of what the motion looks like geometrically. The introduction suggests that a method - I understand this to be the method the paper is describing - should be "rooted in dynamical theory" (line 30). However, the method presented here is purely kinematic, in that it encodes a clever way to find trajectory segments that match a particular geometry. There are some dynamical aspects such as the discussion about limiting anticyclonic frequency ranges as well as inertial and tidal motions, but the "root" of the method is purely kinematic, just like the other methods summarized by Dong et al. (2011). I think it is perfectly okay for the method to be "kinematic" only, but I feel the introduction is overstating or at least mis-stating what the method actually does.

Overarching review questions (from the NPG website) - I have answered them separately in the questionnaire as "no" wherever I had some lingering doubts. Please refer to the list here for details:

* Does the paper contain new and significant results?

The method is "new" only insofar as it is an improvement of an existing method. See also comments above.

* Is the paper of an international standard?

Yes, it is.

* Is the presentation clear and concise?

The presentation is very clear. See comments about length above.

* Does the paper put the obtained results into context, with relevant references?

The paper explains its own methodology well, incl. references to earlier versions and mathematical background, but lacks a bit w.r.t. context with other methods. See comments above.

* Is the length of the paper appropriate?

The paper is very long, but as I wrote above, this is acceptable given that it reads well.

* Is the text fluent and precise?

Yes, the fluency of the text is a strength of this paper.

* Are the title and the abstract pertinent and understandable to a wide audience?

Yes, these are appropriate.

* Are all figures necessary and of appropriate quality?

Comments about length, above, identify two figures that are not absolutely necessary.

The detailed comments below call out a few figure details, in particular a several cases where the figure sizing needs to be adjusted (captions overflow the pages). Some of the thinner lines would display better on my screen if they were either thicker or darker, but this may be an issue with my screen rather than the authors' choices.

Specific comments:

P. 2, ll. 24 ff.: I have glanced over the Dong et al. (2011) reference and the eddy extraction methods listed therein, and am not convinced that applying these to large datasets would be as problematic as the authors here suggest. Firstly, the whole global drifter dataset is a few ten thousand trajectories, which is not prohibitively large. This amount of data can be handled by present-day laptop computers, let alone larger machines. Secondly, at least some of these existing methods identify looping trajectory parts through different mathematical techniques and actually provide computer code to do so. Running existing code over several thousand trajectories does not look problematic at all to me. I agree that the methods I have looked at are not as eloquently described as the one here, and that estimates of errors or significance are lacking. My interpretation is that nobody has tried to run several of these methods side-by-side over a global dataset, not because it is "problematic", but rather because it requires time and money (somebody's salary) to do so.

P. 30, ll. 680 ff.: This paragraph describes a complex situation with multiple oscillatory signals superimposed, and the wavelet spectra (fig. 10) give some insights into what is going on. That said, I am missing the statement that the method looks for elliptical signals, and in this particular situation, the signal just isn't very elliptical. I have looked at altimetry maps from the time and see the cyclonic eddy (the large one discussed here) being pushed around by e.g. an SSH high to the north, which is consistent with what is written here. We cannot expect this to result in pretty, elliptical trajectories (at least if the interaction happens on the same time scales as the eddy rotation), so any detection method will detect all sorts of artefacts when trying to match ellipses. I am not suggesting any changes to the manuscript, but want to reiterate that reality is not

as geometrically perfect as the model we are imposing on it.

Use of wording "geostrophic turbulence" in lines 83, 187, 367: The wording suggests that geostrophic turbulence is some background noise process, and that the eddy signals examined here are something distinguished from this background. Is this really a good interpretation? Effectively, geostrophic turbulence is what fills the oceans with eddies, and if a drifter is trapped inside one of those (as opposed to e.g. a Loop Current Ring), the methodology here should rightfully find it. Drifters might bounce back and forth between multiple eddies from this background flow, in which case any individual eddy event is not detectable here because the trajectories are not looping but rather some random walking. Anyhow, I have stared at surface flow animations on the following website, which show e.g. the Loop Current, but also the open ocean as a "sea of eddies" that I assume is geostrophic turbulence. I would think that the open-ocean eddies would (and should) be part of what this method detects, and feel that "geostrophic turbulence" is part of the signal rather than the noise. Here is the website with the visuals: https://svs.gsfc.nasa.gov/cgi-bin/details.cgi?aid=3827

Minor issues and typos:

Typo, p. 2, l. 24: "Solutions to this problem _has_ been...", should read "have"

Insert space, p. 6, l. 131: "region is left to a sequel._All code developed"

P. 7, footnote 2: Consider moving this footnote into the "Acknowledgments" section.

Insert space, p. 13, l. 294: "univariate signal x(t)_was"

Notation in section 3.1, table A1, and throughout manuscript: I would have preferred the symbol "lambda" for longitude, which is commonly used alongside "phi" for latitude. Change or ignore, at your discretion.

P. 15, l. 360: Incorrect exponent, should be: Omega = 7.292 * 10^(-5) rad s^(-1)

P. 21, footnote 5: Consider moving this footnote into the "Acknowledgments" section.

P. 25, fig. 8a: On my screen, the thin gray lines are hard to see. Recommendation: use more conspicuous colors (e.g. black, dashed lines?).

Typo, p. 28, l. 657: "nonstationary" -> "nonstationarity"

P. 28, fig. 9: I was confused by the black line in panel a, but I suppose this is just the bathymetry? Recommend to remove it, or replace with gray shading, to avoid the association with the black line in panel b.

P. 29, fig. 10: Reduce size to allow caption to fit on page (or consider removing panel a). I would also recommend slightly thicker lines for the colored lines that match fig. 9, just to increase visibility.

Typo, p. 30, l. 696: "It amounts to _a_ idealized...", should read "an"

Typo, p. 32, l. 733: "no effect in improve the stochastic"

P. 33, ll. 760 ff., recommendation: expand a few occurrences of "data" into "real-world data" or similar, just to be extra sure which dataset is being talked about.

P. 33, l. 771, recommendation: remove "more" from "discussed more later"

P. 34, figure 12: Reduce size to allow caption to fit on page.

Typo, p. 36, l. 806: "Thus" -> "This"

P. 36, l. 810: Should "minimum and maxima frequency" be "minimum and maximum frequencies"?

P. 40, fig. 15: Reduce size to allow caption to fit on page.

P. 43, ll. 968/969: The "spheretrans" algorithm is not really "discussed here", apart from the following few sentences (which are appropriate for the purpose). Recommendation: replace "will be briefly discussed here" with "works as follows".

P. 43, l. 980: The GOMED link is missing the "doi.org" part.

P. 43, data availability in ll. 975 ff.: It seems that access to the GOMED dataset (the second of the two links in this paragraph) is restricted and only granted after an application process. This is fine, but the access restrictions on GOMED should be explicitly mentioned in this paragraph (e.g. as requiring registration, non-commercial use only, no derivatives or redistributions allowed). Access restrictions on the trajectory data (the first link) are described properly.

P. 1, l. 11: Following on the previous topic: the abstract claims that the GOMED dataset is made freely available. This is not true and should be corrected, e.g. as "...dataset are made available for non-commercial research use."

P. 4, l. 94: Ditto.
* * *

---

## Referee Comment (RC2) · Anonymous Referee #2 · 20 Oct 2020

The manuscript provides a review of a previously developed method for extracting eddy signal from Lagrangian dataset, and then introduces new developments and an important application to the large Gulf of Mexico drifter data set.

I found the paper interesting, well written and well organized. My only comment is that the summary part of the paper is in my opinion too extended and too detailed. My suggestion is to shorten that part and to highlight more clearly the new developments that address unphysical sign transitions and false positives, introducing statistical significance of the results.

My overall recommendation is that the paper should be published with minor revisions.

---

## Author Comment (AC1) · 13 Dec 2020

We would like to thank the reviewer for their careful reading of our paper and for their thoughtful comments, as these have given us the opportunity to improve the paper. We were particularly happy to read that the reviewer found the length of the paper suitable and that the text flowed well, as we worked hard to accomplish that. The reviewer has raised a number of good points which we will do our best to address. In particular, the reviewer has brought our attention to several statements we have made in the interest of conveying understanding based on personal experience; we have provided more

explanation and backup for these statements.

–Given that a number of methods with similar objectives already exist, it is not really clear why another one is needed. Other methods might not be as elegant or well-documented, but does this alone really justify an entire paper? The case would be clear if it were shown that existing methods deliver wrong or ambiguous results, or as a minimum, if there were a table that showed each method's capabilities and short-comings. The authors mention the existence of earlier methods (e.g. references from Dong et al., 2011), and claim that these methods are problematic to apply to the given dataset. I find this claim unsubstantiated, and possibly incorrect, but have to admit that I have not tried any of the methods myself here. Therefore, this paper is an incremental improvement from the previously existing method by the first author - no more, no less.

The purpose of the paper is not to compare and contrast the performance of various methods. While that would be a valuable undertaking, it is not a trivial exercise. However, we do not believe it to be necessary to prove the superiority of the Lilly and Gascard (2006) method, which this paper builds on, in order to motivate the paper. A primary reason is that, as we have stated in the manuscript, the Lilly and Gascard method now appears to be the method of choice. Because of this, we do not feel it is relevant to try to make generalizations about the plusses and minuses of the various methods proposed or used in the past. It is not the case that another method is in standard use, and we are proposing something different, in which case we would agree that establishing the advantage of a proposed method would indeed be relevant.

It is true in a sense that this paper represents an incremental improvement. However, the ability to assess statistical confidence and eliminate spurious false positives is a significant increment of progress that allows a quantum jump in capability. As seen in Section 4, false positives are quite numerous, and therefore pose a major problem. No other comparable method has attempted to treat the effect of random fluctuations on eddy detections. Consequently, all other methods would be expected to suffer from the effects of false positives.

An important point with regard to this paper is that it explains in detail to the oceano-graphic community, for the first time, a technique that is grounded in ideas from signal processing theory. While the method has previously discussed in the publications in which we developed it, it has not yet been made accessible to the community for which it is intended. In order to accomplish this we have needed to introduce readers to concepts that are mostly likely unfamiliar. This accounts for what we believe is the most valuable contribution of this paper, as well as for its length.

We have modified the Introduction to stress that the intent is to improve an existing method, while limiting the discussion other methods to only a high-level perspective. At the same time, we have sought to clarify the distinction between a trajectory and a signal, as this came up several time in the reviewer's comments. The Introduction now contains the following paragraphs:

"The problem of identifying, and estimating the properties of, coherent eddies in La-grangian trajectories should be distinguished from the problem of describing the aggregate forms of trajectories due to the eddies they contain. To help clarify this, we introduce the term "eddy signal" to mean the displacement of a particle about an eddy's center. We would then see a trajectory as a superposition of different types of signals: e.g., an eddy signal, a near-inertial signal, and a mean flow. Whereas methods such as that of Dong et al. (2011) as well as the spin parameter method of Veneziani et al. (2005a,b) are concerned with identifying or modeling the aggregate trajectory, here we are interested in identifying and extracting eddy signals themselves. This interest complements examinations of the statistical imprint of eddies on trajectories using the spin parameter approach (Griffa et al., 2008; Lumpkin, 2016; Cetina-Heredia et al., 2019)."

"Identifying eddies is trajectories is sometimes equated with finding trajectories that execute loops, and indeed the term "looper" is sometimes used to mean "a trajectory containing an eddy". However, one must be very cautious about forming this equivalence as there is not a one-to-one relationship between trajectory loops and particle displacements due to eddy currents. Simply changing the value of an advecting flow

will alter the appearance of, or even eliminate, trajectory loops for a given eddy signal. Similarly, a trajectory can form a loop for many reasons that do not involve a coherent eddy. For these reasons, eddy detections and property estimates based solely on the visual appearance of trajectories should be considered only as rough approximations. "

"Various methods have been proposed over the years for identifying and extracting eddy signals from Lagrangian trajectories (e.g. Kirwan et al., 1984, 1988; Armi et al., 1989; Flament et al., 2001; Testor and Gascard, 2003; Lankhorst, 2006); see Dong et al. (2011) for a useful review. Generally speaking, a difficulty faced by such methods is that fact that the frequency of the eddy signal is not only unknown a priori, it also tends to vary substantially with time, as seen for example in Flament et al. (2001). This frequency modulation makes the study of eddy signals substantially more difficult than, for example, studying tides, which occur at known and fixed frequencies. Narrowband methods such as band- passing or complex demodulation would therefore perform quite poorly. In order to accommodate such frequency modulation, the above methods generally contain free parameters that must be chosen by an analyst. Because of the need for hand tuning for individual trajectories, applications of such methods to large datasets would be problematic."

"A major step in the eddy extraction problem was taken by Lilly and Gascard (2006). In that paper, an innovative and powerful method from signal analysis termed wavelet ridge analysis (Delprat et al., 1992; Mallat, 1999) was modified for application to Lagrangian trajectories. That method is designed to detect and analyze modulated oscillations, that is, oscillations whose amplitude and frequency vary as a function of time. This type of signal accords well with our physical expectations for the trajectory of a particle trapped in a vortex. The wavelet ridge method is able to automatically extract frequency-modulated signals occurring somewhere within a specified frequency band, without the need to tune parameters for an individual timeseries. A compelling aspect of this method is that it begins with the specification of the type of object we are looking for, namely, a modulated oscillation, a type of signal for which there exists a solid theoretical foundation (Gabor, 1946; Picinbono, 1997). "

– The paper purposefully keeps the oceanographic results to a minimum, and announces a future manuscript to follow up. This is appropriate for a methodology paper, but a bit more context should be provided: Have other Lagrangian eddy detection methods re- ported the same asymmetry between cyclones and anticyclones, or is this asymmetry a new finding or specific to the Gulf of Mexico? Are similar eddy properties, incl. the asymmetries between cyclones and anticyclones, generally present in eddy-resolving numerical circulation models?

We have added the following paragraph near the end of Section 4.7 to address this question.

"Cyclone / anticyclone asymmetries at the mesoscale are expected on theoretical grounds (Matsuura and Yamagata, 1982; Cushman-Roisin and Tang, 1990; Arai and Yamagata, 1994; Cho and Polvani, 1996), and have also been reported in observations in studies that apply the spin parameter method to surface drifters (Griffa et al., 2008; Lumpkin, 2016). The details of this asymmetry would be expected to vary regionally. In our Gulf of Mexico eddy census, the occurrence of intense mesoscale cyclones that are much smaller than the anticyclones is perhaps not surprising, since at least some of the cyclones are believed to form from instabilities on the periphery of the Loop Current or Loop Current eddies. Further investigation of the reasons behind the observed asymmetry, and the extent to which it generalizes to other parts of the ocean, would be a promising topic for future investigation. "

– There is a large number of self-citations in the references. While it is a good thing to provide background information, a malevolent interpretation could be that not many people other than the authors themselves care about this topic. This makes the paper appear less significant. As of Oct. 19, 2020, there were no public discussion comments about the online version of the paper. This, too, reflects poorly on the significance of

the paper. Perhaps the authors can still ask a handfull of peers to submit comments before the deadline? This could at least show that interested readers exist.

The self citations reflect the fact that progress in this problem has been made largely by the author and collaborators. In our judgement, the self citations and number of community comments are not reliable indicators of overall interest. A better indicator might be the total number of views. As of the initial draft of this response (3 Nov), this manuscript had 498 total views. For comparison, the ten papers submitted before this one (and which thus have had more time to accumulate views) that are still under review currently have 377, 374, 465, 371, 451, 368, 256, 364, 366, and 334 views. This indicates that the level of interest in our work is in fact very high.

–In the introduction section, the word "dynamic" is used in a way that does not match my expectations. I was expecting "dynamics" to refer to explanations (forces, energy bud- gets, Navier-Stokes equations) behind the observed motions, and would distinguish this from "kinematics", the mere description of what the motion looks like geometrically. The introduction suggests that a method - I understand this to be the method the paper is describing - should be "rooted in dynamical theory" (line 30). However, the method presented here is purely kinematic, in that it encodes a clever way to find trajectory segments that match a particular geometry. There are some dynamical aspects such as the discussion about limiting anticyclonic frequency ranges as well as inertial and tidal motions, but the "root" of the method is purely kinematic, just like the other methods summarized by Dong et al. (2011). I think it is perfectly okay for the method to be "kinematic" only, but I feel the introduction is overstating or at least misstating what the method actually does.

This is a good point, and we have removed the statement in question from the Introduction. It is true that the model is essentially kinematic, however, it is a kinematic model that aligns very well with dynamical solutions of elliptical eddies; but this is not really addressed in the present paper. The distinction we had been trying to make vs. other methods has been better clarified by discriminating those based on trajectory forms vs.

those based on eddy signals, as in the paragraphs quoted above.

–P. 2, ll. 24 ff.: I have glanced over the Dong et al. (2011) reference and the eddy extraction methods listed therein, and am not convinced that applying these to large datasets would be as problematic as the authors here suggest. Firstly, the whole global drifter dataset is a few ten thousand trajectories, which is not prohibitively large. This amount of data can be handled by present-day laptop computers, let alone larger machines. Secondly, at least some of these existing methods identify looping trajectory parts through different mathematical techniques and actually provide computer code to do so. Running existing code over several thousand trajectories does not look problematic at all to me. I agree that the methods I have looked at are not as eloquently described as the one here, and that estimates of errors or significance are lacking. My interpretation is that nobody has tried to run several of these methods side-by-side over a global dataset, not because it is "problematic", but rather because it requires time and money (somebody's salary) to do so.

The problem is not one of computational power, it is of the need for hand-tuning arising from the existence of frequency modulation in the signals of interest. This was admittedly not properly explained in our earlier draft. We hope that the revised paragraphs in the Introduction quoted above will satisfy the reviewer's concerns. The statement that the methods are problematic to apply to very large dataset is not, we believe, one that would be controversial to those who have used them.

–P. 30, ll. 680 ff.: This paragraph describes a complex situation with multiple oscillatory signals superimposed, and the wavelet spectra (fig. —10) give some insights into what is going on. That said, I am missing the statement that the method looks for elliptical signals, and in this particular situation, the signal just isn't very elliptical. I have looked at altimetry maps from the time and see the cyclonic eddy (the large one discussed here) being pushed around by e.g. an SSH high to the north, which is consistent with what is written here. We cannot expect this to result in pretty, elliptical trajectories (at least if the interaction happens on the same time scales as the eddy rotation), so any

detection method will detect all sorts of artefacts when trying to match ellipses. I am not suggesting any changes to the manuscript, but want to reiterate that reality is not as geometrically perfect as the model we are imposing on it.

This comment somewhat blurs the distinction between a signal and a trajectory. As we have tried to clarify above, it is not the trajectory that modeled as being elliptical, it is a signal within that trajectory. The squarish forms of the trajectory during the second half of the record are actually quite consistent with an eddy advecting another eddy and match the ellipse-plus-residual model very well. However, we do agree with the reviewer that in the first part of the record, this model leads to a structured residual, suggesting model misfit. We have now expanded this discussion as follows:

"This complex trajectory is a good example of a situation in which the multiplicity $M(t)$—the apparent number of modulated oscillations at any moment—is greater than one. The presence of inertial oscillations superposed on a background mesoscale eddy, which accounts for the cusps seen in Fig. 9a, is fairly common in this dataset. The superposition of two lower-frequency ridges, seen during yeardays 60–110, occurs less frequently. A physical hypothesis consistent with these two ridges is that a particle is trapped in an eddy that is itself being advected by another eddy, with the lower-frequency signal arising from advection on the exterior flank of the second eddy. The superposition of these two signals accounts for the "wobble" in Fig. 9a, where the center of the tight loops in the middle of the plot appears to vary over time. "

"During the second half of the record, the sum of three oscillations—two eddy-like signals and an inertial signal—accounts for most of the variability apart from an over northward drift. This indicates that the unobserved components model of Eq. (39) can generate quite complex and irregular trajectories. During the first half, however, the residual curve exhibits more irregular oscillations, suggesting that during this time period the variability is less well matched to our proposed model. The more irregular behavior of the oscillation that dominates during this time period suggests a particle that is weakly trapped on the flanks of the eddy rather than within its solid-body core."

–Use of wording "geostrophic turbulence" in lines 83, 187, 367: The wording suggests that geostrophic turbulence is some background noise process, and that the eddy signals examined here are something distinguished from this background. Is this really a good interpretation? Effectively, geostrophic turbulence is what fills the oceans with eddies, and if a drifter is trapped inside one of those (as opposed to e.g. a Loop Current Ring), the methodology here should rightfully find it. Drifters might bounce back and forth between multiple eddies from this background flow, in which case any individual eddy event is not detectable here because the trajectories are not looping but rather some random walking. Anyhow, I have stared at surface flow animations on the following website, which show e.g. the Loop Current, but also the open ocean as a "sea of eddies" that I assume is geostrophic turbulence. I would think that the open-ocean eddies would (and should) be part of what this method detects, and feel that "geostrophic turbulence" is part of the signal rather than the noise. Here is the website with the visuals: https://svs.gsfc.nasa.gov/cgi-bin/details.cgi?aid=3827

The conceptualization of geostrophic turbulence as a kind of background process is precisely what we are proposing, and this interpretation is strongly supported by our previous work in Lilly et al. (2017). We have explained this more throughly now in the following two paragraphs in the Introduction:

"The problem of false positives can be understood as follows. In previous work (Lilly et al., 2017), we have shown that Lagrangian trajectories in forced-dissipative quasi-geostrophic turbulence can be usefully separated into two classes, those that contain high-frequency oscillatory signals associated with trapping within coherent eddies, and those that do not. Trajectories in the latter class are remarkably similar to those resulting from a type of damped random walk, see Figs. 2 and 3 therein. This supports the conceptual model of trajectories containing eddies consisting of the sum of an eddy signal, superposed on a stochastic process that arises from geostrophic turbulence and that may be considered "noise" from the point of view of detecting eddies."

"The stochastic background flow can be the source of spurious features that masquerade as coherent eddies, which can be understood as follows. In the one-dimensional case, a discrete random walk is intuitively described as a drunk staggering between lampposts. In the two-dimensional case, the drunk has a grid of lampposts available for their staggering. From time to time the drunk will, by chance, happen to turn in a circle, or oscillate back and forth between two lampposts. This illustrates why, in applying the wavelet ridge analysis to timeseries of stochastic processes analogous to the random walk, oscillatory events are occasionally detected. One would not wish to confuse random features arising from the turbulent background with the organized oscillations due to coherent eddies."

The reviewer also suggests, with regard to the animation, that geostrophic turbulence fills the ocean with eddies and that these should be detectable more or less everywhere (if we understand their point correctly). This is not the case, and reflects the subtlety involved in shifting between Eulerian and Lagrangian views. The visual appearance of an eddy-like feature in an Eulerian frame does not necessarily correspond to a long-lived oscillatory signal in the Lagrangian frame. One may see instantaneously closed streamlines that lead to locally curved trajectories, but not to multiple orbits as would result from a long-lived potential vorticity anomaly. It is the latter that we are seeking to identify.

Minor isues:

We have corrected all of the typos the reviewer spotted—thank you for these—and considered their other suggestions. Other comments we respond to individually as follows.

–P. 25, fig. 8a: On my screen, the thin gray lines are hard to see. Recommendation: use more conspicuous colors (e.g. black, dashed lines?).

We have thickened these lines and explained that one reason that they are hard to see is that they over overplotted by other lines.

–P. 28, fig. 9: I was confused by the black line in panel a, but I suppose this is just the bathymetry? Recommend to remove it, or replace with gray shading, to avoid the association with the black line in panel b.

Thank you for pointing this out. We have updated the figure and added this line to the caption: "The dark gray shading is the continent and the thick light gray line is the 500 m isobath, as in Fig. 1."

–P. 29, fig. 10: Reduce size to allow caption to fit on page (or consider removing panel a). I would also recommend slightly thicker lines for the colored lines that match fig. 9, just to increase visibility.

We have done so, as with the two other figures the reviewer makes this comment on, and also note that at these sizes, the captions will fit on the page in the format used for the published version. We have slightly increased the line thickness, as recommended, and also ensured that the thin white contours do not overplot the colored lines.

–P. 33, l. 771, recommendation: remove "more" from "discussed more later"

We have taken the reviewer's suggestion.

–P. 43, ll. 968/969: The "spheretrans" algorithm is not really "discussed here", apart from the following few sentences (which are appropriate for the purpose). Recommendation: replace "will be briefly discussed here" with "works as follows".

We have taken the reviewer's suggestion.

–P. 43, data availability in ll. 975 ff.: It seems that access to the GOMED dataset (the second of the two links in this paragraph) is restricted and only granted after an application process. This is fine, but the access restrictions on GOMED should be explicitly mentioned in this paragraph (e.g. as requiring registration, non-commercial use only, no derivatives or redistributions allowed).

This is a good point, and we thank the reviewer for mentioning it. We have added the

following sentence: "In keeping with conditions stipulated by the funding agent, this dataset is made freely available for academic use with the agreement that it shall not be sold, profited from, or redistributed."

–P. 1, l. 11: Following on the previous topic: the abstract claims that the GOMED dataset is made freely available. This is not true and should be corrected, e.g. as "...dataset are made available for non-commercial research use."

–P. 4, l. 94: Ditto.

We have added the word "noncommercial" at both locations.

---

## Author Comment (AC2) · 13 Dec 2020

We would like to thank the reviewer for their time in reading our paper and for their comments. The only comment was that the summary was too detailed, and that this should be shortened to highlight the new contributions. In re-reading the paper and looking for places to shorten, we found we could remove several technical paragraphs from the Introduction, and added more high-level discussion that places the work in context. We hope that this will address the reviewer's concern.

---

## Author Response (AR1)

The reponses to the reviewers have been detailed in our public responses to their comments. In particular, we have made the following changes, with line numbers referring to the resubmitted manuscript:

1. Revised lines 24--56 of the Introduction in order to better clarify the relation of this method to others, addressing a concern of Reviewer 1
2. Removed several more technical paragraphs from the Introduction in order to address a comment made by Reviewer 2
3. Discussed further the physical reasoning for modeling geostrophic turbulence as a stochastic process, lines 72--84 of the Introduction, in response to a question by Reviewer 1
4. Modified the discussion of the sample eddy, lines 670--683, in response to a question from Reviewer 1
5. Added additional physical discussion of the cyclone/anticyclone assymetry, lines 924--931, in response to a question from Reviewer 1
6. Corrected some typos, and added minor clarifications throughout, including those brought up by the reviewers
7. Made minor changes to notation (e.g. swapping underscore stars for overbars, renaming varsigma to xi, renaming nu to v)
8. Added several new average and meta-data quantities to the dataset, and to the list in Table A1
9. Improved and simplified the definition of our significance parameter. This has led to a number of text changes in Section 4.6, and a new version of figure 14, but no noticable changes to the other plots. The reason for this change is explained on lines 860--865.
10. Reduced the number of figures by one by absorbing the previous figure 15 into the right column of the new figure 12, and adding a comparable column in figure 13.